# Catching Up of Latecomer Economies in ICT for Sustainable Development: An Analysis Based on Technology Life Cycle Using Patent Data

**Na Zhang \*, Chao Sun, Min Xu, Xuemei Wang and Jia Deng**

School of Maritime Economics and Management, Dalian Maritime University, Dalian 116026, China; sun1998126@126.com (C.S.); minxu0916@163.com (M.X.); wangxm0104@163.com (X.W.); djia0326@163.com (J.D.)
\* Correspondence: nazhang@dlmu.edu.cn; Tel.: +86-15041120993

**Abstract:** In the digital economy era, ICT plays a vital role in supporting the sustainable and high-quality development of latecomer economies. Using technology life cycle analysis and patent data from the United States Patent and Trademark Office (1960–2014), this study analyzed the catching-up characteristics of latecomer economies (with the US as the first mover), including take-off time, growth time, growth rate, and ceiling values in nine sub-fields of information and communication technology (ICT). We applied the logistic and bi-logistic model to reveal the sequence of technological development and growth speed of different economies in different ICT sub-fields. The results show that European economies (Great Britain, France, and Germany) and the US developed first, followed by Japan, Korea, and Taiwan, with China (Mainland) coming later; Asian economies (Japan, Korea, Taiwan, and China (Mainland)) displayed synchronous development strategies, while European economies displayed non-synchronous development strategies. Asian economies are catching up with the US, whereas European economies are standing still both in imitation and indigenous processes. Korea and Taiwan prioritized catching up with a few sub-fields in the indigenous process. Finally, we analyzed the technological convergence among economies in their catching-up processes and proposed policy implications for the sustainable development of ICT latecomers.

**Keywords:** catching up; latecomer economies; ICT; sustainable development; technology life cycle

## 1. Introduction

Driven by the new round of scientific and industrial revolution, economies worldwide are accelerating into the digital economy era. Technological iteration and upgrades in ICT play vital roles in supporting the development of the digital economy in various economies [1–4]. Governments have always aimed to be at the leading edge of ICT development and innovation. However, few studies have focused on the development process and trajectory of ICT in various economies to analyze the development stages and strategies of different economies. Therefore, from the perspective of the technology life cycle, this study applies an S-curve model to compare and analyze the technology life cycle characteristics and catching-up strategies of latecomer economies in nine sub-fields of ICT, in order to provide theoretical support and policy implications for the sustainable development of various economies.

Many economists have constructed macro- and microeconomic models to discuss the economic growth and catching up of latecomer economies. The earliest studies originated with the economic growth model proposed by Solow in the late 1950s and the early 1960s, which was used to explain the long-term stable growth mechanism of an economy [5]. The study found that technological progress was the main driving force of economic growth. The Solow residual provides a method for measuring the contribution of technological progress to economic growth and lays the foundation for the study of total factor productivity (TFP). Based on Solow's model, Romer and Lucas proposed endogenous growth

theory models in the late 1980s and the early 1990s, respectively. Romer [6] introduced knowledge as an endogenous factor in economic growth and emphasized the internal generation of technological progress and the driving role of innovation. Lucas [7] proposed the role of human capital accumulation in economic growth. On this basis, Aghion and Howitt [8] proposed a creative destruction model that emphasized the driving effect of technological progress on economic growth and explained how technological catching up can be achieved through the innovation and destruction of existing technologies. Similarly, Jones [9] discussed an endogenous growth model based on R&D, emphasized the role of technological innovation in economic growth, and analyzed the relationship between technological catching up and R&D investment.

Based on the above theoretical research, some scholars have tested the convergence of economic growth rates among different regions or countries and the factors influencing economic catching up through empirical research. The economist Robert Barro proposed the Barro Convergence Model [10,11]. The model suggests that the economic growth rate of rich regions may slow, while that of poor regions may accelerate, thus achieving economic convergence. Despite the trend towards convergence, the actual rate of convergence may be influenced by political stability, the level of education, and market openness. Comin and Hobijin's [12] empirical study showed that the most important determinants of a country's rate of technology adoption are the country's human capital endowment, type of government, openness to trade, and adoption of the previous generation of technology. As these variables converge across countries, the overall rate of diffusion increases significantly. Aghion and Howitt [13] tested the important issues of Schumpeterian growth theory using micro-data, including innovation policies, intellectual property protection, education, and R&D incentives, to reveal the driving factors and policy impacts of technological innovation and catching up.

These studies mention the importance of technological progress, technological innovation, R&D investment, and the absorption of advanced technology in economic growth and economic convergence. In recent years, several studies have investigated technological capability gaps and catching-up characteristics among different economies from the perspectives of technology and innovation management. Lee and Lim [14] built a model of technology and market catching up to explain the different technological evolutions of selected industries in Korea. They determined the conditions for catching up to occur. Hu [15] demonstrated that producers in Korea and Taiwan built innovation capabilities by creating complementary knowledge for Japanese firms in thin film transistor liquid crystal displays, and latecomers sought to expand production by selecting certain technological fields. Lee and Malerba [16], Shin [17], Kang and Song [18], and Li et al. [19] adopted a catching-up cycle framework and identified windows of opportunity (technology, demand, or policy) that may emerge during successive changes in a sector. Szczygielski et al. [20] found that government aid for R&D activities contributed to the catching up of technology followers. Cimoli et al. [21] presented a North–South technology gap model that combines the Schumpeterian approach to technical and structural change with the Keynesian perspective on effective demand and the Balance-of-Payments (BOP) constraint as drivers of growth.

However, there are three limitations to the current studies on latecomers' catching up among economies. First, most studies examine technological catching up by building theoretical frameworks or case studies, and few long-term studies have examined latecomers' technological catching up in ICT from a longitudinal perspective based on technological life cycle analysis. Existing studies on technological catching up in ICT are conducted using short-term analysis and lack dynamic time-series analysis, especially comparing timeline differences and technological growth speed.

Second, most studies have focused on a single specific sub-field of ICT in which latecomers have been successful in catching up. Although latecomers—particularly Korea and Taiwan—have achieved remarkable technological development (such as semiconductor and consumer electronics), this fact does not hold, nor does it have the same meaning in all

technological sectors. Thus, a broad comparative perspective is required across sub-fields and economies to assess ICT technology trajectories.

Third, some studies have discussed the convergence or divergence of economic growth rates among economies and analyzed their driving factors. However, few studies have analyzed the convergence of technological growth rates among latecomers and first movers in different ICT sub-fields. This limits the possibility of revealing the technological gaps and driving factors of technological convergence.

Technological development is a dynamic and cumulative process by which latecomers expand their knowledge base in both scope and depth [22–26]. Latecomers' catching up is a learning process that includes imitations, adoption, and assimilation of mature technologies from first movers [27,28]. Technology life cycle (TLC) analysis is an important method for analyzing the technology levels of different economies from long-term and dynamic perspectives [29,30]. TLC analysis can map the entire process along the technology development trajectory, including the key time points of each stage and growth rate, which helps reveal the entire process of the technological catching-up cycle from a dynamic perspective. As the catching up of latecomers includes two processes—imitation and indigenous innovation—we depict catching up using two TLC curves.

Thus, this study conducts a longitudinal study on ICT using double TLC curves to examine the catching-up characteristics of latecomer economies, including the take-off time, growth time, growth rate, and ceiling values, using the US as the first mover (benchmark). Furthermore, we examine technological convergence in different ICT sub-fields among different economies and driving factors. We used USPTO-granted patent data from 1960 to 2014 and applied logistic and bi-logistic models to reveal the sequence of technological developments of different economies in different ICT sub-fields and the length of the period during which such sub-fields are sustained and found out the driving forces of technological convergence among latecomers and the first mover for the sustainable development of catching-up economies in ICT.

The remainder of this paper is organized as follows. Section 2 defines technological catching up and ICT development background in terms of catching up. Section 3 presents the methodology and data (i.e., the S-curve model, data sources, and ICT sub-fields). Section 4 demonstrates the overall characteristics of the TLCs of economies, including the take-off time, growth rate, and growth time. Section 5 presents the technological convergence analysis, and Section 6 concludes the paper.

## 2. Catching Up and ICT Development Background

### 2.1. Conceptualizing Catching Up

The term *catching up* has been used in macroeconomics to analyze the extent to which the growth of income or productivity of different economies (significantly behind the frontier at the start of the period) enables them to catch up with leading economies (first movers) by the end of the period [31]. Thus, there is a narrowing gap in productivity or income between the first mover and the latecomer. This study also describes this as the process by which a latecomer increases its technological capability (technological catching up) vis-à-vis the first mover. We propose that if knowledge is codified and freely available, latecomer economies will grow faster than first movers because the former will benefit from existing technologies developed by the latter at a lower cost and more rapid pace, thereby reducing the gap between the two [32]. In this regard, we define catching-up economies as those that generate more rapid technological innovation than first movers [33].

Many late-industrializing economies have attempted to create technological innovation to achieve technological growth, despite the unfavorable innovation environment. While most have not progressed much, a few latecomers have shown rapid technological catch-up. In other words, there are falling-behind and standing-still economies. Based on the definition of catching up and the growth rate, falling behind denotes economies with a lower technology growth rate than the first movers, whereas standing still denotes economies with the same growth rate as the first movers. Catching-up economies tend to



achieve higher productivity in technological sectors with shorter cycle times and easier access to knowledge, whereas first movers show the opposite.

One common observation from previous research is that catching up is a long process that passes through multiple stages. Some studies proposed that latecomers' learning processes start with the accumulation of technological capabilities from reverse engineering (i.e., imitation) of foreign technologies to indigenous innovation [34,35]. The paradigm shift from imitation to indigenous innovation necessitates the separation of production into two development cycles to capture two different growth processes [14,19,36]. In the imitation process, latecomers emphasize technological accumulation, during which they develop basic technological capabilities by improving technologies acquired through technology transfers or spillovers from inward FDI [14,22]. In the indigenous innovation process, latecomers can achieve sustained catching up globally when they consider the global market a springboard for acquiring further strategic assets, especially critical technologies, brands, and human resources [14]. With indigenous technological capability building, latecomers can narrow the gap with the first movers in technological innovation and market share in the global market and sustain their initial catching up, as the technology frontier is a moving target that can be easily moved away from latecomers, leading their early catching up to no avail [14,22].

## 2.2. Catching Up in ICT

ICT has become an important technological and strategic field for governments and has attracted significant attention from policy makers, managers, and researchers [37]. This study focuses on ICT for several reasons. First, ICT is an important driver of economic growth. Research on G20, G7, and the Organization for Economic Co-operation and Development (OECD) member countries has shown that ICT has become a new engine of economic growth in developed economies [1,38,39] and created opportunities for emerging Asian economies, including Korea, China, and Singapore [40].

Second, ICT has attracted considerable attention. In recent decades, emerging Asian economies have implemented policies to promote ICT development during their catching-up periods. For instance, Japan introduced successive national strategies, including the e-Japan Strategy I in 2001, the e-Japan Strategy II in 2003, the u-Japan Initiative in 2004, i-Japan 2015 in 2009, and the Fifth and Sixth Basic Plans for Science and Technology (2016–2020) in 2016. Korea released "IT839" in 2006 to emphasize the development of eight services, three infrastructures, and nine growth engines, which was followed by the "IT future strategy in Korea 2009" (focusing on software, broadcast communication, and broadband network) and "RFID Promotion strategy" in 2011. China took information technology (IT) as one of seven strategic emerging industries in its "Twelfth Five-year Plan" in 2010 and included the digital economy in government reports for five consecutive years since 2017.

Third, ICT has become enabling technology. ICT has boomed in the information age. ICT-based technologies have also enabled the research, development, and growth of technologies in many other fields, such as applied science, engineering, energy, health, and transport. Economies that do not catch up in ICT also fall behind in other ICT-related technologies. As such, findings in the ICT field might translate into other ICT-enabled technologies as well.

In the history of ICT development, the US first introduced numerous radical innovations (including computing, mobile telecommunications, robots, and computer software). Catching up began in the 1970s when some European countries followed the US in some sub-fields. Subsequently, Japan dominated (especially in the field of semiconductors and consumer electronics), followed by Korea and Taiwan. Since then, China, India, and Brazil have grown rapidly.

## 3. Methodology and Data

### 3.1. Technology Life Cycle and S-Curve

The notion of the TLC was first introduced by Little [41]. Since then, most studies on TLC analysis have been based on the assumption that a technology or a group of technologies has a cycle from emerging and growth to maturity and recession [42]. Following this assumption, many scholars and practitioners have attempted to predict the future course of a technology using past and present information to chart S-curved or double S-curved progressions [43,44]. Figure 1 shows the S-curve of the TLC in four stages. As stated above, the catching-up processes of latecomers can be reflected as two cycles of trajectories: the imitation process and the indigenous process. Thus, we used a double S-curve to depict the TLC characteristics.

The horizontal axis of the S-curve represents time, and the vertical axis shows the cumulative performance, as shown in Figure 1. In this study, cumulative performance is defined as cumulative technological capability. Based on Schumpeter's theory of innovation, technological capability can serve as a cause for catching up. Technological capability is the ability of a firm (economy) to transform an idea into a new one with economic value, which is determined as a function of both technological effort and the existing knowledge base [14]. Different from technical efficiency, which is defined as the effectiveness with which a given set of inputs is used to produce an output, technological capability can reflect technological catching-up characteristics more directly.

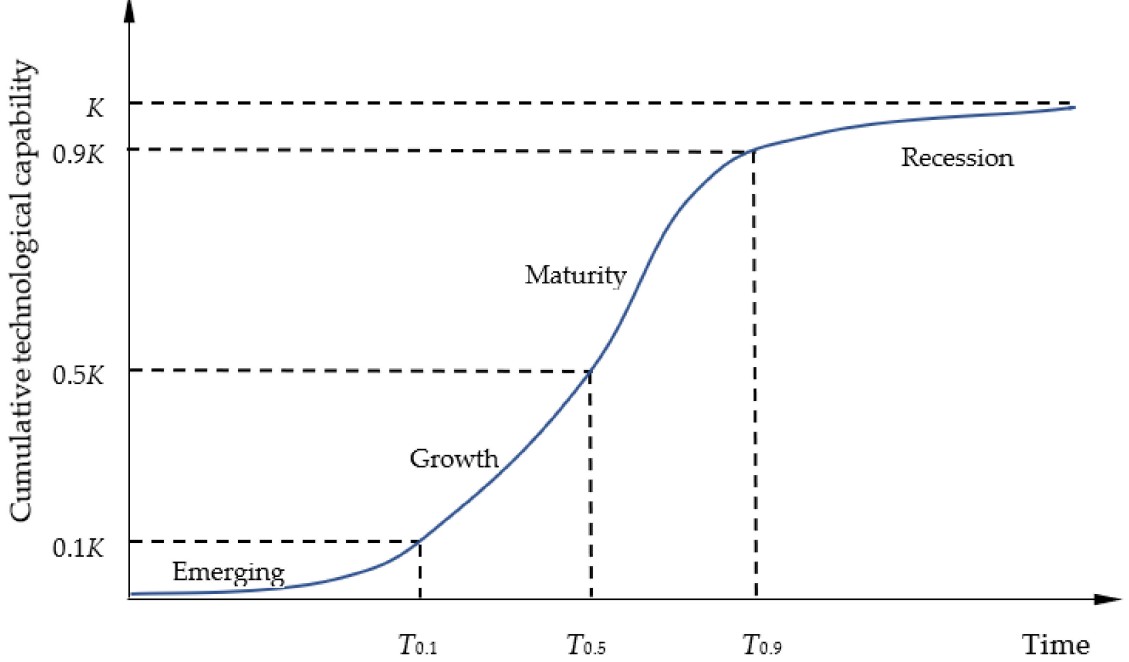

**Figure 1.** S-curve of TLC.

### 3.2. Logistic and Bi-Logistic Model

Various growth curves have been used to represent technology life cycles, such as logistics, Gompertz, and Lotka–Volterra competition models, to obtain estimates of the technology's prospects [25,45–47]. The logistic model is representative of a symmetrical S-curve, whereas the Gompertz model produces an S-curve that increases more sharply but begins to taper earlier than the logistic model [48]. The Lotka–Volterra competition model explains multigenerational technology advancement, in which multiple new technologies continue to develop in a particular technology area and then spread. The logistic model originates in the biological realm and is often applied to model the production of technology,

given its effectiveness in capturing the changing nature of technological growth [49]. Therefore, we chose the following simple logistic model:

$$Y(t) = \frac{K}{1 + \exp(-bt + c)} \tag{1}$$

where $Y(t)$ is the cumulative stock at time $t$, $K$ is the ceiling value, $b$ is a coefficient depicting the growth rate, and $c$ is a constant that positions the curve on a timescale.

The breakpoints of four stages are indicated by $0K$, $0.1K$, $0.5K$, and $0.9K$ (emergence, growth, maturity, and recession). $T_{0.1}$ is the starting time of the growth stage, and $T_{0.9}$ is the starting time of the recession stage [45]. The equations for $T_{0.1}$ and $T_{0.9}$ are as follows:

$$T_{0.1} = \frac{c - \ln 9}{b} \tag{2}$$

$$T_{0.9} = \frac{c + \ln 9}{b} \tag{3}$$

The growth time ($T_{0.1-0.9}$) is defined as the time of the growth and maturity stages, where the ceiling value starts from $0.1K$ and reaches $0.9K$. The formula for $T_{0.1-0.9}$ is as follows:

$$T_{0.1-0.9} = T_{0.9} - T_{0.1} = \frac{\ln 81}{b} \tag{4}$$

To describe the overall characteristics of the TLC, we used three indicators: ceiling value ($K$), growth rate ($b$), and starting time of the growth stage. The ceiling value represents the carrying capacity, the growth rate represents the maximum development speed, and the starting time of the growth stage represents the take-off time of the technology [43,50].

Assuming that catching up includes a shift from imitation to indigenous innovation, each process can be modeled using a simple logistic model. Thus, the bi-logistic model is suitable for modeling catching-up processes. The bi-logistic model is as follows [36]:

$$Y(t) = Y_1(t) + Y_2(t) = \frac{K_1}{1 + \exp(-dT + e)} + \frac{K_2}{1 + \exp(-fT + g)} \tag{5}$$

where $K_1$ and $K_2$ are the ceiling values for the first and second processes, respectively; $d$ and $f$ are the growth rates; and $e$ and $g$ are constants.

In this study, we first used a simple logistic model to obtain the overall growth rate of the sample economies, compared it with the growth rate of first movers, and depicted their catching-up (or falling-behind) characteristics. After that, we used a bi-logistic model to compare the growth rate and growth time among latecomers in two catching-up processes to find out the specific sub-fields and processes in which each economy is catching up (or falling behind). Table 1 summarizes these indicators. The indicators of $K$, $0.9K$, $0.5K$, $0.1K$, $T_{0.1}$, $T_{0.5}$, and $T_{0.9}$ are labeled in Figure 1.

We used computer software called Loglet Lab 4 (https://phe.rockefeller.edu/LogletLab/, accessed on 1 July 2021) to calculate the parameters of the nonlinear logistic model. Comparing with Shazam, MATLAB, and SPSS to manipulate the logistic curve, Loglet Lab 4 operates the data with hundreds of iterations based on a Monte Carlo simulation, making the results highly reliable [45].

### 3.3. Data

Patent analysis is considered one of the most established, directly available, and historically reliable methods for quantifying the output of a technology system [51]. The advantages of using patent data are as follows. First, they are a direct outcome of the inventive process and, more specifically, of those inventions that are expected to have a commercial impact. Patent data are appropriate for capturing the proprietary and competitive dimensions of technological change. Second, because obtaining patent protection

is time-consuming and costly, it is likely that applications are filed for inventions that, on average, are expected to provide benefits that outweigh these costs. Third, patents are broken down by technical field and thus provide information not only on the rate of inventive activity but also on its direction. Fourth, patent statistics are available in large numbers and for long time-series. Although patent-based indicators have limitations, ample research has demonstrated their validity as proxies for measuring technological capabilities.

**Table 1.** Summary of all indicators.

| Symbol | Indicator | Meaning of Indicator |
|:---:|:---:|:---:|
| $K$ | Ceiling value | Carrying capacity |
| $b$ | Growth rate | Maximum development speed |
| $T_{0.1}$ | Starting time of the growth stage | Take-off time of the technology |
| $T_{0.5}$ | Starting time of the maturity stage | |
| $T_{0.9}$ | Starting time of the recession stage | |
| $T_{0.1-0.9}$ | The time of the growth stage and maturity stage | Growth time |
| $K_1$ | Ceiling values for the first process | |
| $K_2$ | Ceiling values for the second process | |
| $d$ | Growth rate for the first process | |
| $f$ | Growth rate for the second process | |

Within the scope of the S-curve model, the most direct approach is to use the accumulated number of patents as technology develops to fit the growth curves [48,52,53]. It has been found that a cumulative stock of patents over time generally follows an S-shaped curve [53,54]. For example, Anderson [43] fitted logistic growth functions to US patent stocks during 1920–1990, and Liu and Wang [45] applied S-curve models (logistics and Gompertz) to predict the development trend of the biped robot walking technique. Hence, patent data can be used as an index for measuring technological capability [25,29,50].

However, patent counts from different patent offices are not always comparable because of the different patent policies, patenting costs, approval requirements, and enforcement rules for patenting in different economies. A common remedy is to use patent data from a single patent-granting country, such as the US, to standardize the differences between patent offices and the unit of innovation, making cross-country comparisons possible [36]. In this study, we considered only USPTO grant data.

The reasons for choosing the USPTO include the following. (1) As the US represents the largest and the most technologically advanced market in the world, any sufficiently large invention patented anywhere with a global market in mind is likely to be patented in the United States as well [25,33]. (2) A single patent-granting database is necessary to ensure that technological capability across different economies is comparable [25].

We examined catching-up processes over a long period (the past 50–60 years). Because there were two- or three-year time lags from patent publication to grants, we only measured granted patents with publication years between 1960 and 2014. Thus, the cumulative granted patent stock is the dependent variable ($Y(t)$), and the publication year (1960–2014) is the independent variable ($t$). We tagged the publication years from 1 to 55 to ensure a more accurate evaluation of the coefficients.

Among the many ICT-related technology classifications, we considered nine growth drivers defined in the IT839 Strategy of the Korean Ministry of Information and Communication [55]. These nine growth drivers are expected to lead to future industrial growth in the future. The nine sub-fields and their USPC concordances are presented in Table 2. This classification was commonly used by Lee, Kim, and Park [56] and certified by Lee, Lee, and Yoon [57]. Therefore, we used this technological classification.

**Table 2.** ICT sub-fields and USPC concordance.

| ICT Sub-Field | Abbreviation | USPC Classes |
|---|---|---|
| Mobile telecom/telematics | MOB | 340, 375, 379, 701 |
| Broadband/home network | NET | 370 |
| Digital television/broadcasting | DTV | 345, 348, 349, 353, 367, 381, 382, 386 |
| Computing/auxiliaries | COM | 235, 361, 365, 700, 708, 710, 713, 714, 719 |
| Intelligent robot | ROB | 318, 706 |
| Radio frequency identification ubiquitous sensor network | RFID | 342, 343, 455 |
| Information technology system on chip/united parts | SOC | 438, 711, 716 |
| Embedded software | ESW | 341, 712 |
| Digital content/software solutions | SOL | 705, 707, 715, 717 |

We calculated the total number of ICT patents granted by the USPTO (1960–2014) in all economies worldwide. In the US (1,116,179), Japan (345,924), Korea (78,293), Germany (50,094), Great Britain (32,531), France (27,257), Taiwan (24,239), and China (Mainland) (15,935) the number of patents was greater than 15,000, whereas the number in other economies was less than 6000. As these eight economies have the highest ICT technological capacity, we considered them sample economies (the US, Japan, Korea, Germany, Great Britain, France, Taiwan, and China (Mainland)). The US is regarded as the first mover, and we analyzed the catching-up characteristics of three European economies (Great Britain, Germany, and France) and four Asian economies (Japan, Korea, Taiwan, and China (Mainland)).

Figure 2 shows the number of patents granted to the eight economies in the nine sub-fields. According to Figure 2, there are significant gaps in the number of patents among the economies. The US and Japan have more patents than other economies. The cumulative patent stock in the US is 1,116,179—three times larger than Japan's (345,924); thus, they are dominant in ICT. Korea, Germany, Great Britain, and France are next, whereas Taiwan and China (Mainland) have the smallest numbers of patents. There are also large gaps in patent numbers among different sub-fields; COM and DTV are relatively larger than the other sub-fields, followed by MOB and SOC; ESW has the smallest patent number.

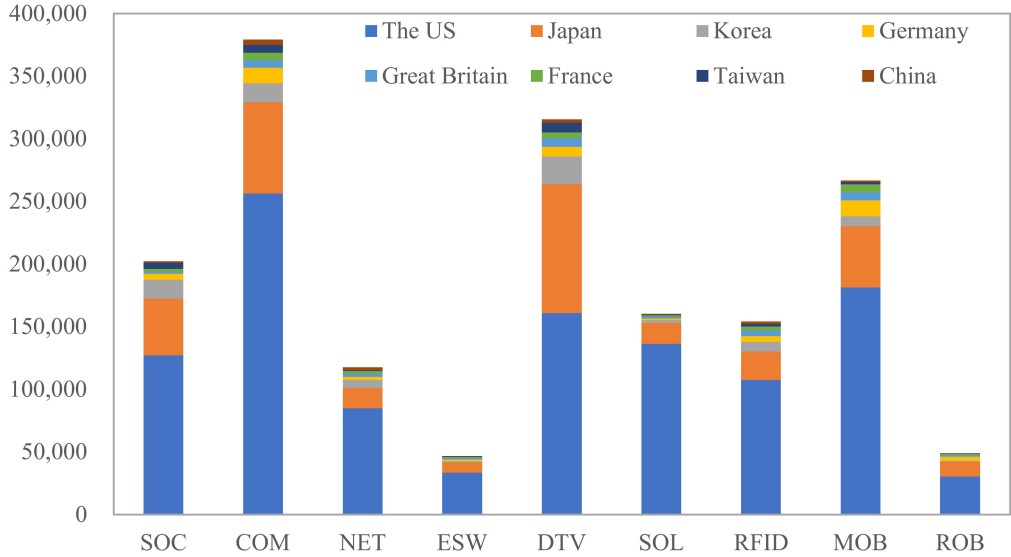

**Figure 2.** The patent numbers of eight economies in nine sub-fields.

## 4. Results Analysis Based on Technology Life Cycle

We fitted a simple logistic model (Equation (1)) and bi-logistic model (Equation (5)) to all nine sub-fields of the eight economies. The t-values of all the coefficients in the logistic model were statistically significant.

### 4.1. Overall Catching Up Comparison Based on a Simple Logistic Model

We calculated the take-off time, growth rate and time, and ceiling value of the TLC using a simple logistic model (see Appendix A, Tables A1–A3). Figures 3–11 show the overall characteristics of the TLC for the eight economies in each sub-field. The horizontal coordinate denotes the take-off time of the economies, and the vertical coordinate denotes the growth rate. Cumulative patent numbers (1960–2014) and ceiling values are in parentheses.

As shown in Figures 3–11 the take-off times in Great Britain, France, Germany, and the US are earlier than those in Asian economies (Japan, Korea, China (Mainland), and Taiwan) in all sub-fields. Among Asian economies, Japan was the earliest economy, taking off in the 1990s, followed by Korea and Taiwan in the 2000s, and China (Mainland) was the latest economy in 2010. The standard deviations of the take-off time of the nine sub-fields for each economy were also calculated (last column in Appendix A, Table A1), which denotes the time differences in the take-off time of different sub-fields. Great Britain (12.80) had the largest standard deviation, followed by the US (11.79), France (11.42), and Germany (9.57). The standard deviations of Western economies (i.e., the US, Great Britain, France, and Germany) are more significant than those of Asian economies—they adopt a non-synchronous development strategy. However, Asian economies (i.e., Japan, Korea, China (Mainland), and Taiwan) develop nine sub-fields nearly simultaneously—they adopt a synchronous development strategy.

The growth rates of Asian economies (Japan, Korea, Taiwan, and China (Mainland)) are over 10%. Japan has a faster growth rate than Western economies; Korea and Taiwan have higher growth rates than Japan, and China (Mainland) has the fastest growth rate—nearly twice that of Korea and Taiwan. These economies accumulate technological capabilities through adoption, assimilation, and rapid learning from mature foreign technologies and grow quickly to catch up with Western economies.

These four Asian economies are typical catching-up economies, as we defined catching-up economies as those whose patent growth rates are higher than those of the first mover (the US). However, it is noteworthy that these economies still have technology gaps in terms of current accumulative patent numbers and ceiling values compared to the US, indicating low technological vitality due to insufficient accumulation at the emerging and growth stages.

By contrast, European economies have the same growth rate as the US in COM, ESW, DTV, RFID, MOB, and ROB, indicating that they are standing still in these six sub-fields. They had a lower growth rate than the US in terms of SOC, NET, and SOL, indicating that they fell behind in these three sub-fields. There is no catching-up sub-field in European economies.

Combining the take-off time and growth rate in a simple logistic model (see Figures 3–11), the catching-up pattern presents a flying geese paradigm [58–60]. This paradigm postulates that Asian economies will catch up with Western economies as part of the regional hierarchy. The underdeveloped economies in the region could be considered to be "aligned successively behind the advanced economies in the order of their different stages of growth in a wild geese-flying pattern". The leading geese in this pattern are Japan, a second-tier economy consisting of South Korea and Taiwan, followed by China (Mainland). It may be argued that China (Mainland) is currently benefiting from flying geese patterns, as it takes up manufacturing activities that were formerly undertaken in Japan, Korea, and Taiwan.

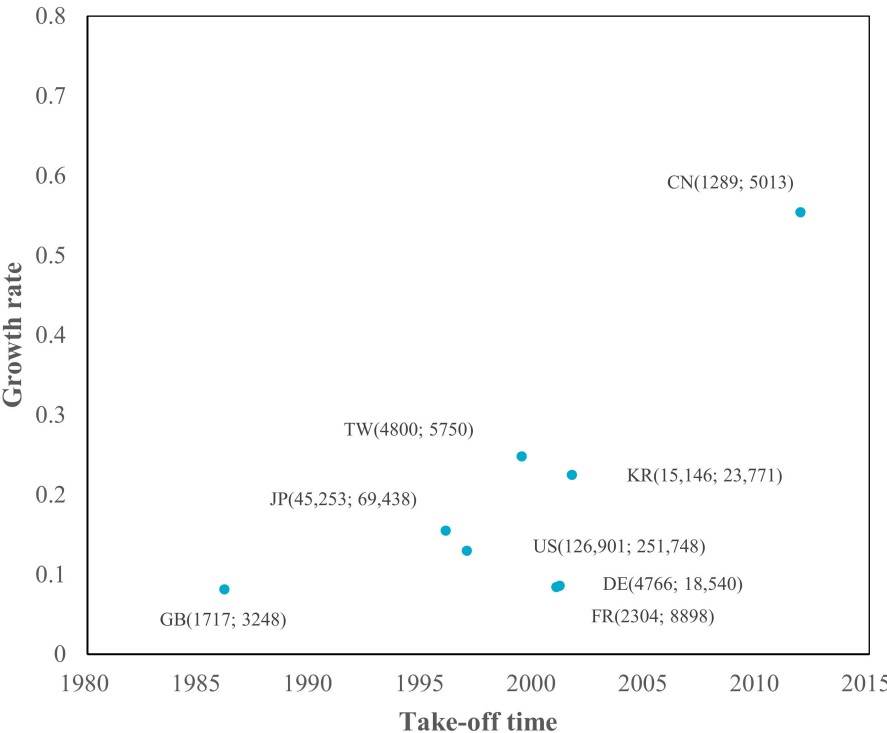

**Figure 3.** TLC characteristics in SOC. Note: US, FR, GB, DE, JP, KR, CN, and TW are short for the United States, France, Great Britain, Germany, Japan, Korea, China (Mainland), and Taiwan, respectively.

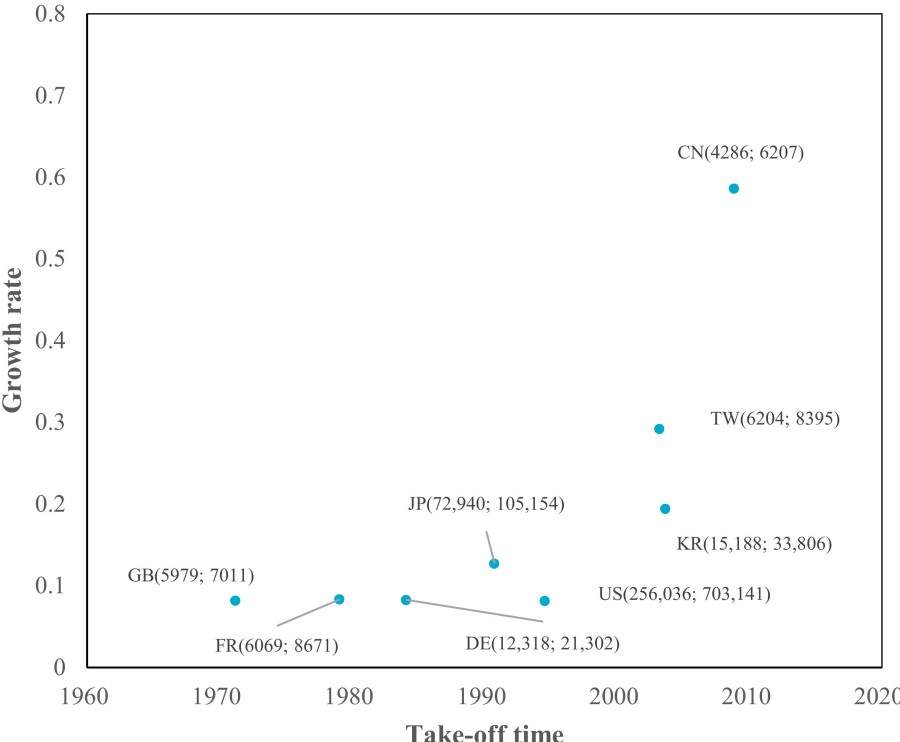

**Figure 4.** TLC characteristics in COM.

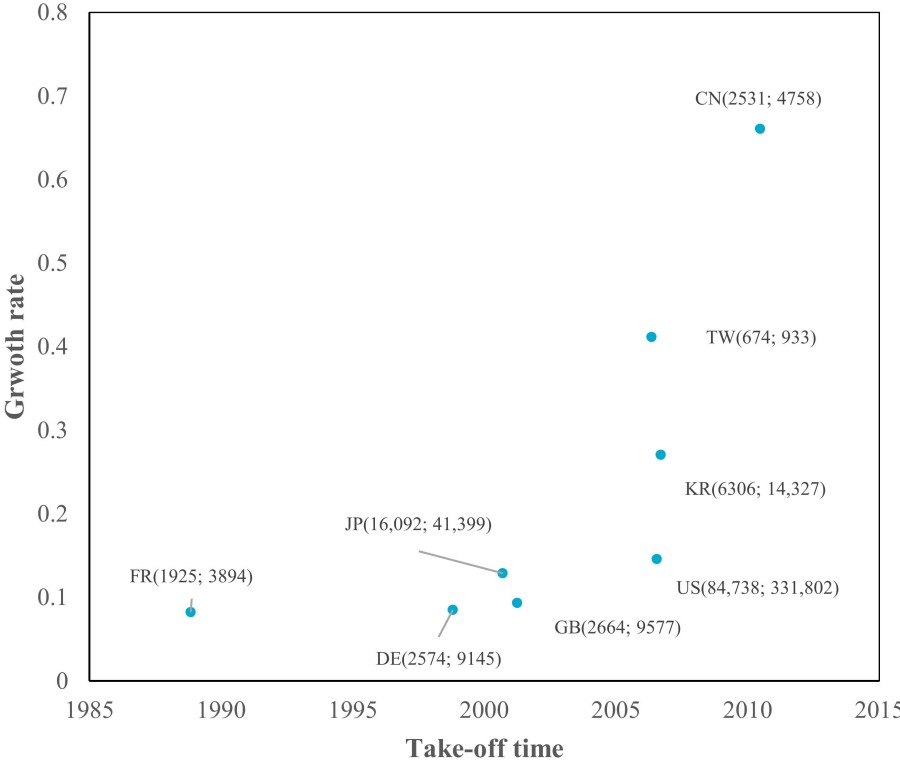

**Figure 5.** TLC characteristics in NET.

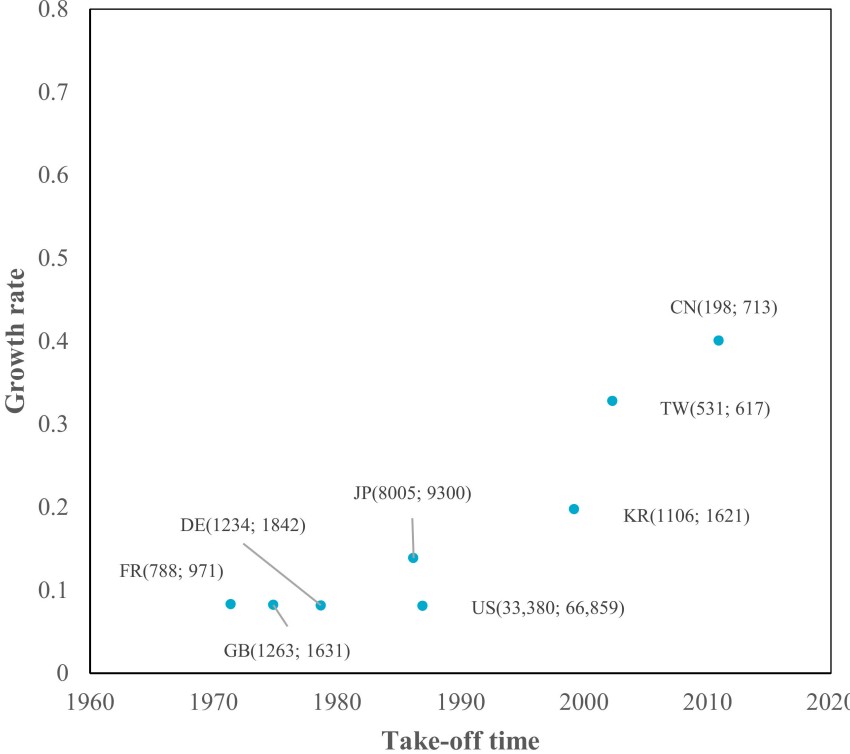

**Figure 6.** TLC characteristics in ESW.

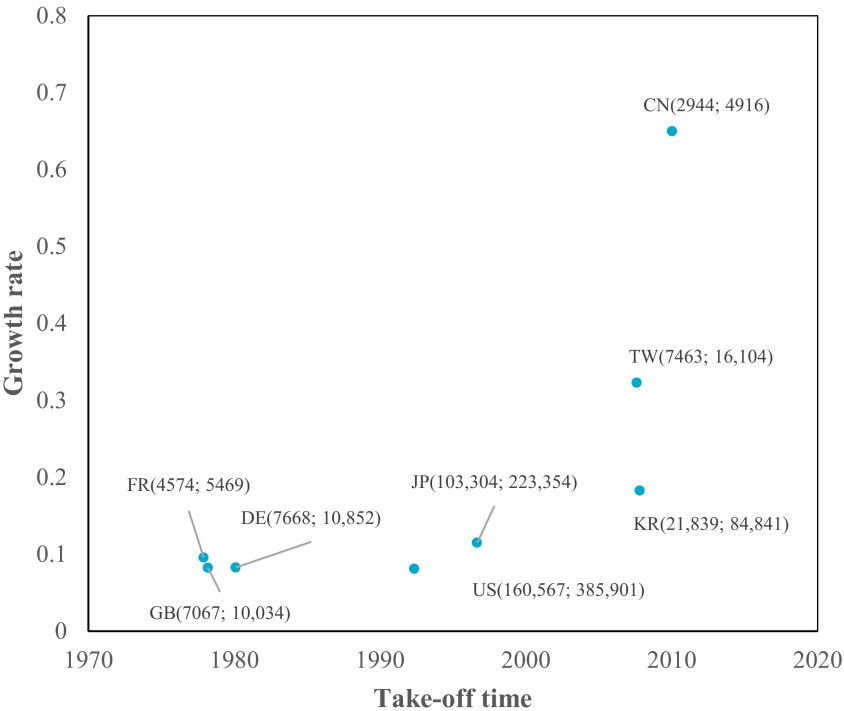

**Figure 7.** TLC characteristics in DTV.

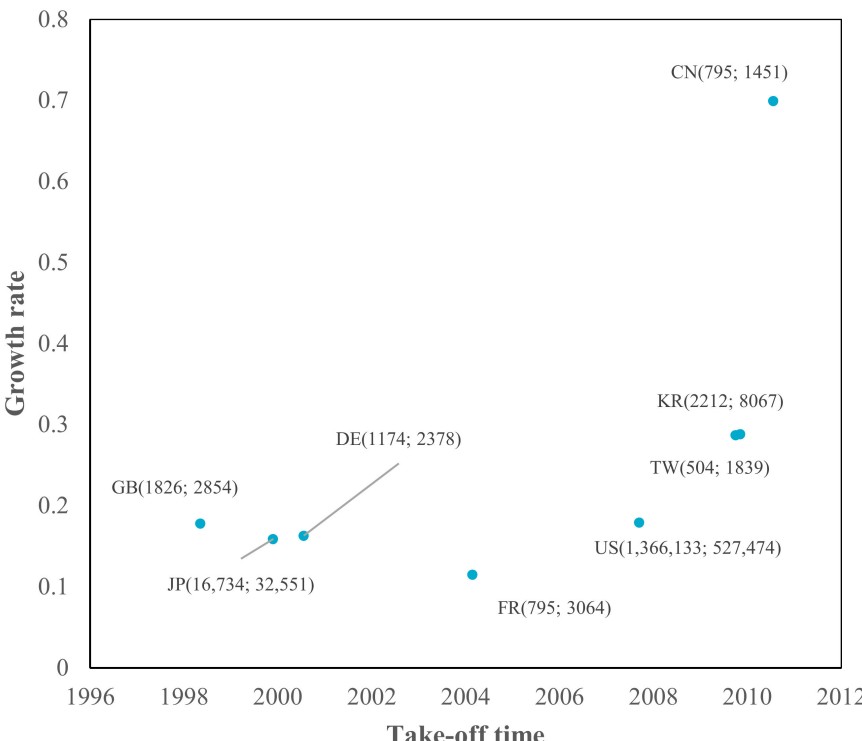

**Figure 8.** TLC characteristics in SOL.

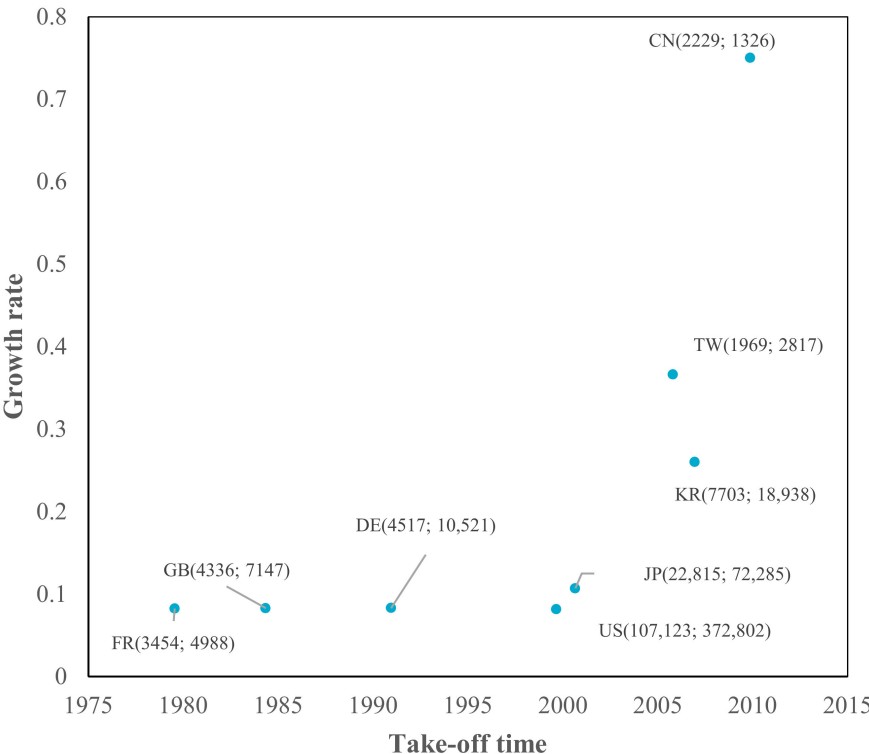

**Figure 9.** TLC characteristics in RFID.

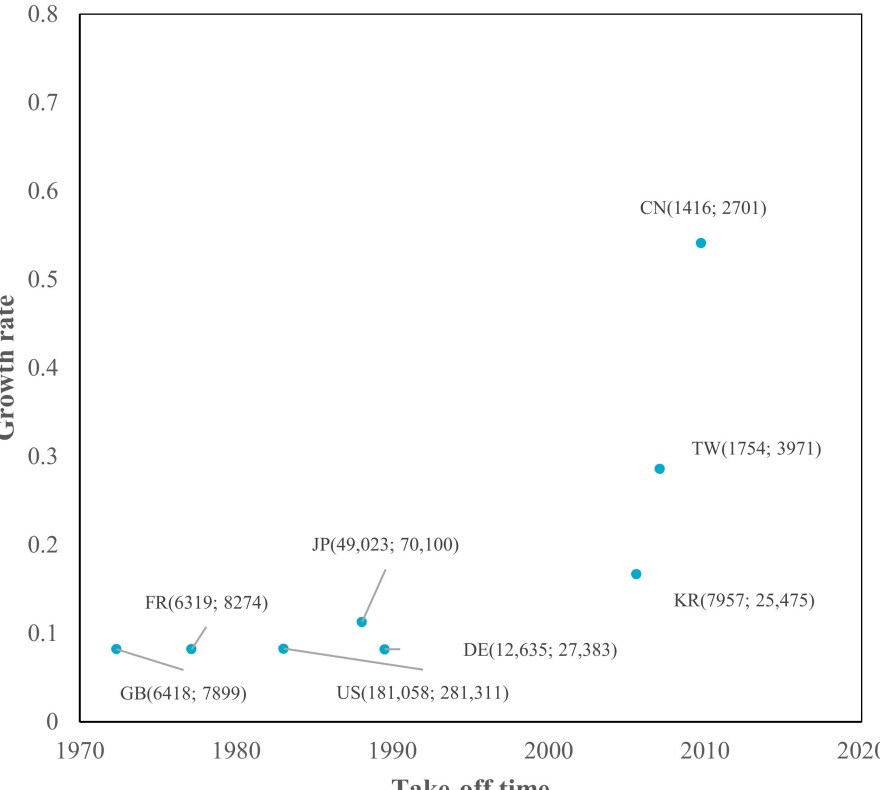

**Figure 10.** TLC characteristics in MOB.

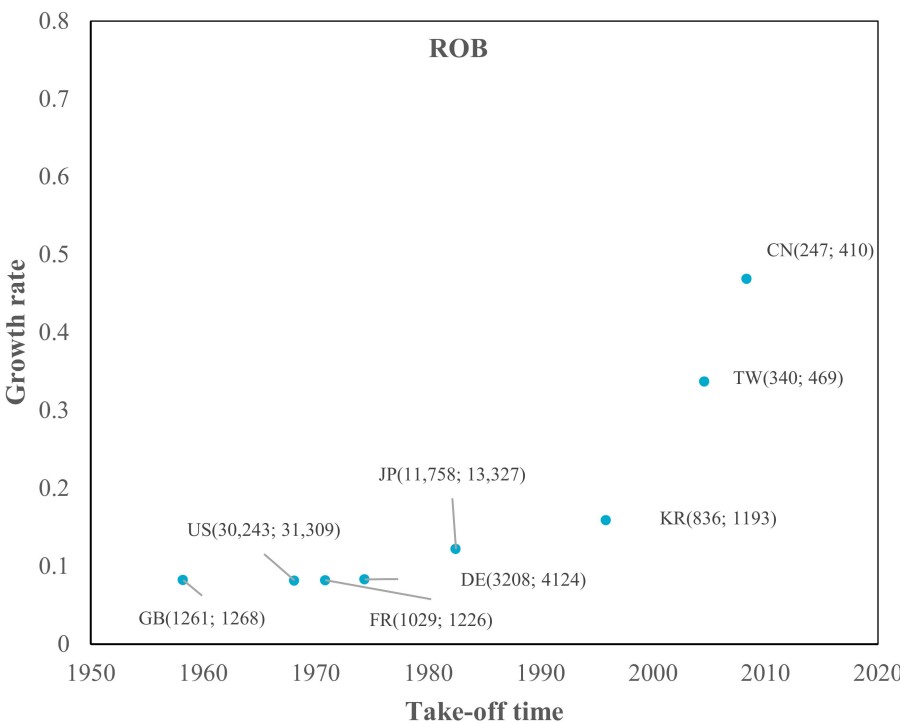

**Figure 11.** TLC characteristics in ROB.

### 4.2. Analysis of Two Catching-Up Processes Based on a bi-Logistic Model

Using the bi-logistic model, we calculated the take-off time, growth rate, time, and ceiling value of two curves for two catching-up processes among seven economies (except the US) in nine sub-fields (see Appendix B, Tables A4–A6). Figures 12–20 show the TLC characteristics of the first and second curves for the seven economies in each sub-field separately. The horizontal axis denotes the take-off time of the first and second curves, and the vertical axis denotes the growth rate of the first and second curves.

We observed distinct patterns in China (Mainland), Korea, and Taiwan. Their take-off time for both curves (imitation and indigenous innovation process) is considerably later than that of the other economies. High growth rates in the first phase reflect rapid imitation. In the second curve (indigenous innovation process), the growth rate of China (Mainland) displays a dramatic increase, which is more than 1 in eight sub-fields (except for ESW, with a growth rate of 0.95, which is slightly less than 1). As higher growth rates imply shorter growth times, the observations suggest that China (Mainland) has difficulty achieving sustainable development because of insufficient accumulation in their imitation stage.

Korea and Taiwan showed dramatic increases in the growth rates of DTV and SOL, which were twice the growth rate of the first curve. In addition, Korea showed an increase in NET and RFID, and Taiwan showed a rise in MOB. For Asian economies, catching up occurs not only in the imitation process but also in the indigenous innovation process. However, although they have higher growth rates than the US, they have shorter growth times. Thus, they must pay more attention to sustainable growth in DTV, SOL, NET, and RFID in Korea, and DTV, SOL, and MOB in Taiwan.

The pattern for Japan is more similar to that of European economies: earlier take-off times (with Great Britain being slightly earlier in the imitation stage) and lower growth rates that do not decrease or increase dramatically, such as China (Mainland), Korea, and Taiwan. Notably, the three European economies are catching up in SOL, with an increased growth rate from the first to the second process. Germany increased its growth rate from the first to the second process in four sub-fields (SOC, NET, ESW, and RFID), whereas Great Britain increased only SOC and NET, and France increased MOB, indicating more positive development trends. The three European economies are standing still in other sub-fields.

However, compared to China (Mainland), Korea, and Taiwan, European economies have a longer growth time in both in the first and second processes, presenting sustainable growth.

From the imitation process to the indigenous innovation process, China (Mainland) showed a significant increase in its growth rate in all nine sub-fields. Korea and Taiwan have priority strategies in choosing prior sub-fields for their indigenous innovation processes. Owing to technology fusion and interaction between different sub-fields, advantageous development in a few sub-fields could bring about development in other sub-fields, leading to development in the overall domain. For European economies, there was no significant increase in growth rate from the first to the second process. Considering that it is difficult to develop all sub-fields simultaneously, Europe may need to select priority sub-fields for development. Sub-fields that exhibit higher growth rates in indigenous processes may be preferred.

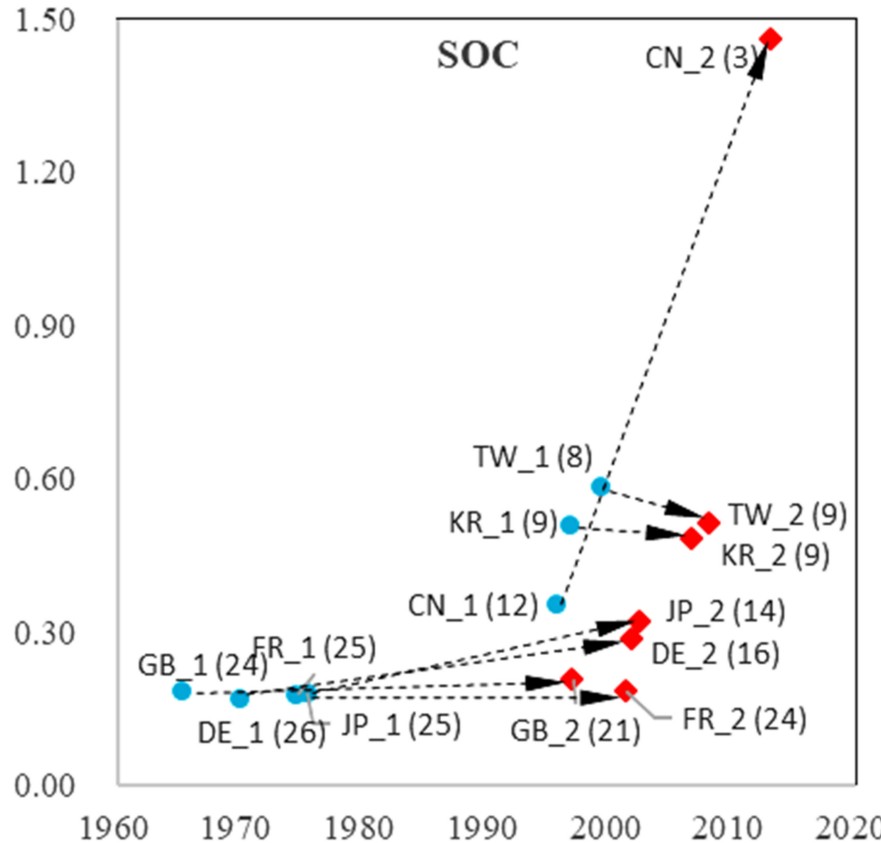

**Figure 12.** TLC characteristics in the two curves in SOC. Note: The first and second curves are represented by blue circles and red diamonds, respectively; growth times are in parentheses. The dotted arrows represent the shift from the first to the second curve.

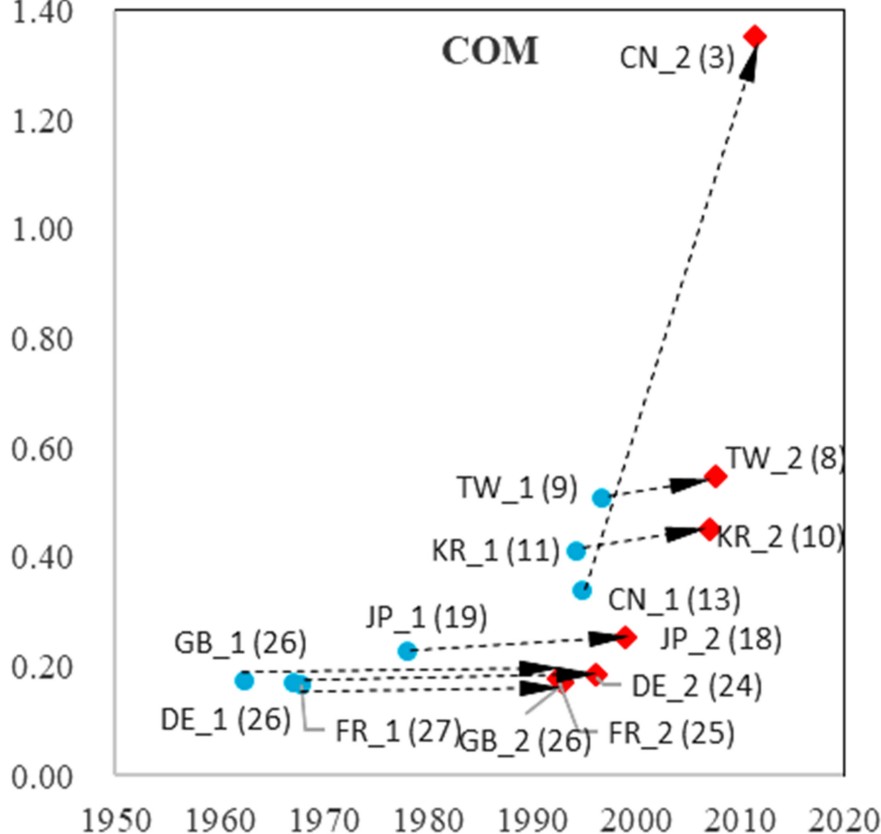

**Figure 13.** TLC characteristics in two curves in COM.

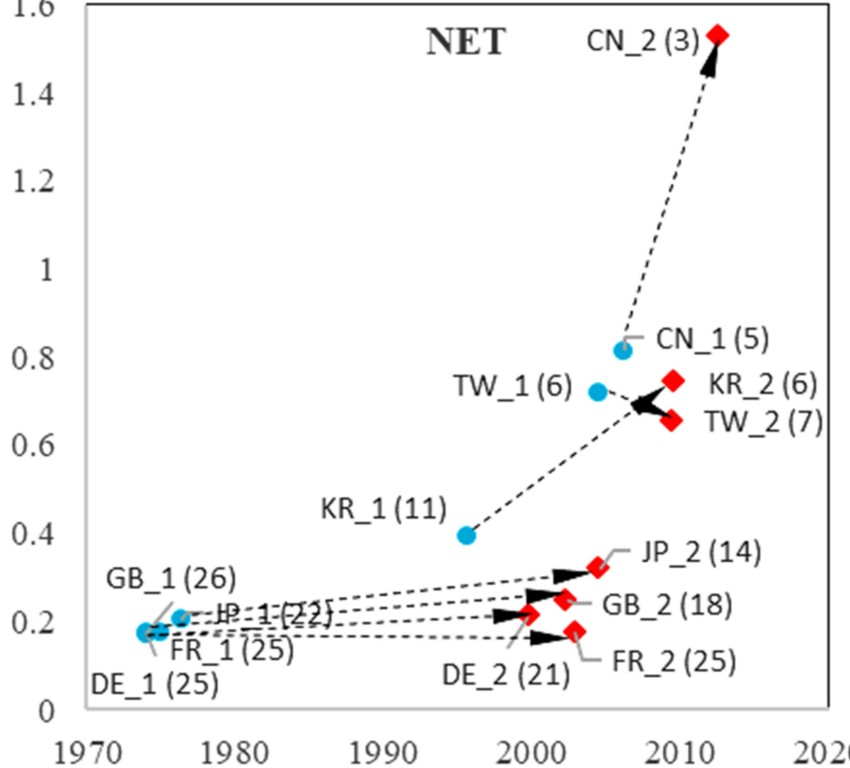

**Figure 14.** TLC characteristics in the two curves in NET.

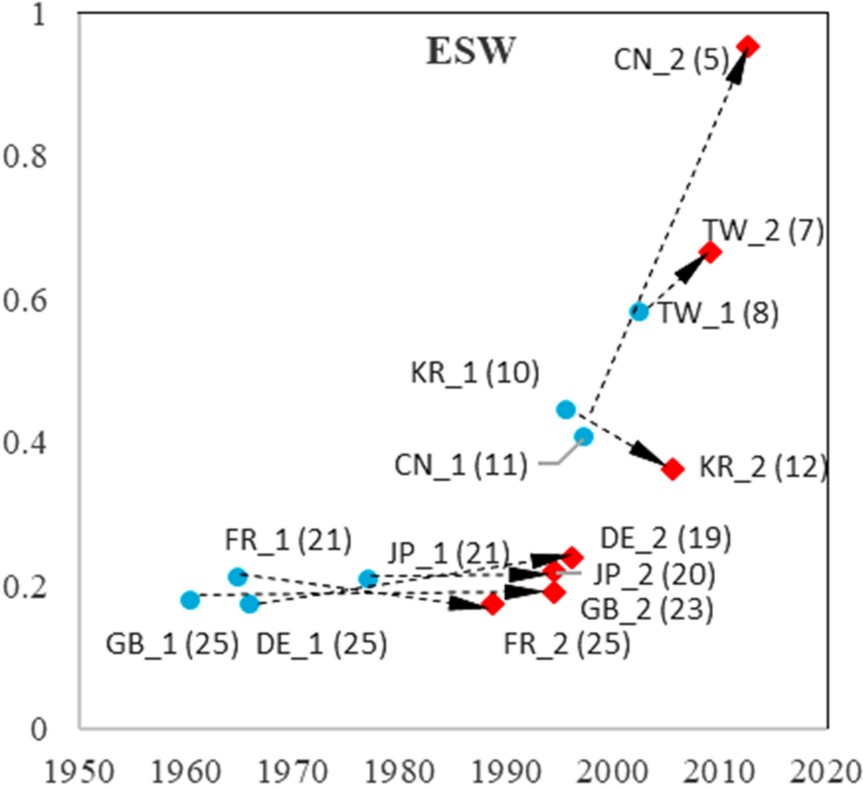

**Figure 15.** TLC characteristics in two curves in ESW.

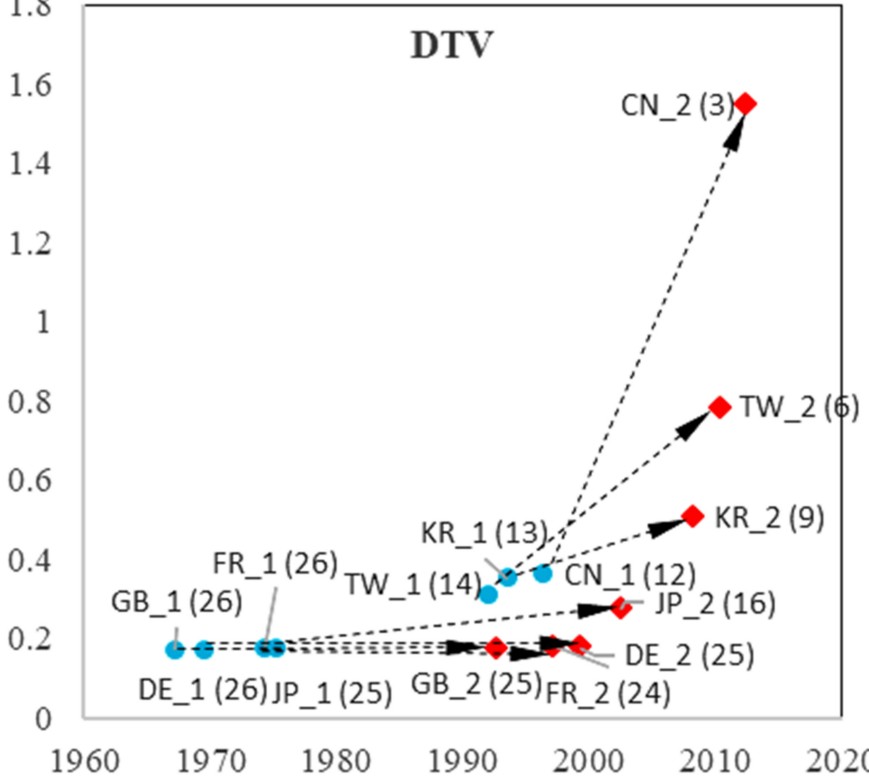

**Figure 16.** TLC characteristics in the two curves in DTV.

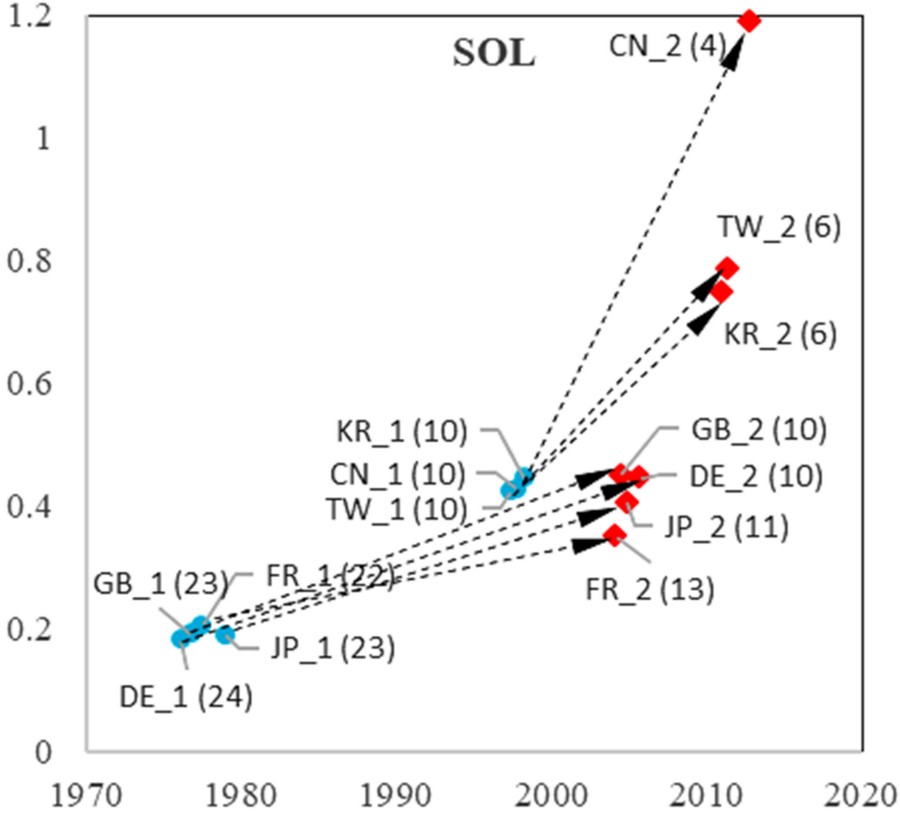

**Figure 17.** TLC characteristics in two curves in SOL.

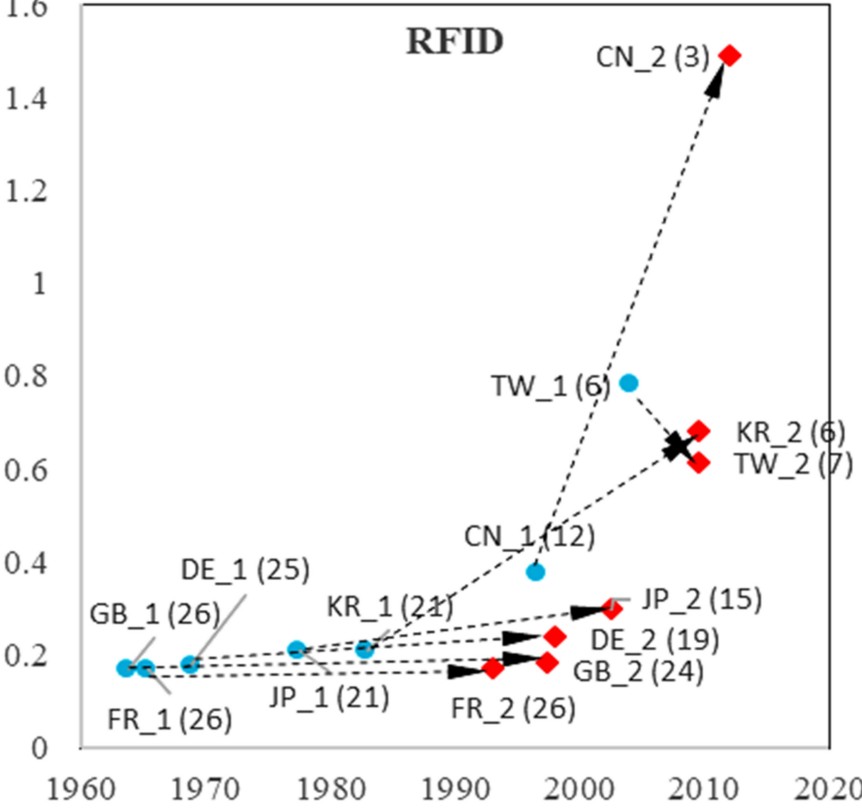

**Figure 18.** TLC characteristics in the two curves in RFID.

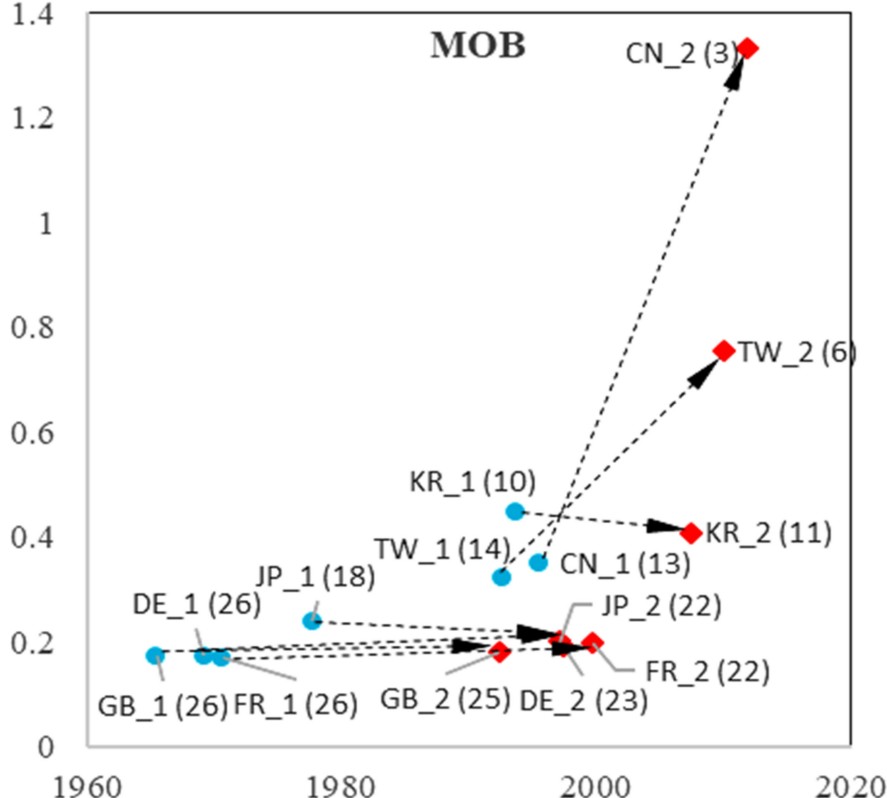

**Figure 19.** TLC characteristics in two curves in MOB.

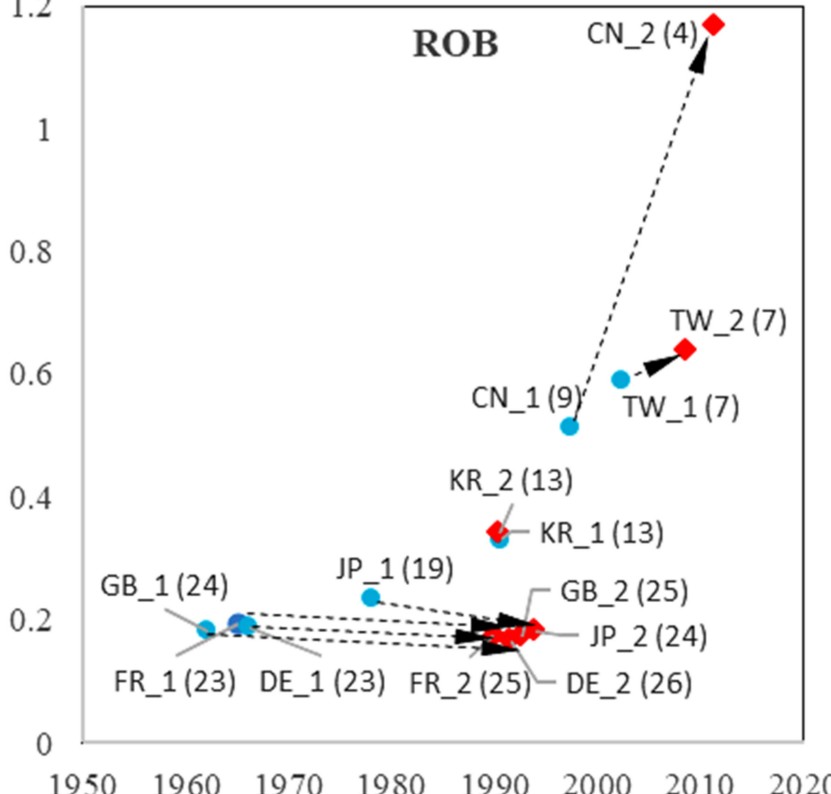

**Figure 20.** TLC characteristics in the two curves in ROB.

## 5. Technological Convergence Analysis

The above research makes a preliminary analysis of the technological catching up of different ICT economies through the technology life cycle model and shows that different economies show differences in the process of technological catching up in the long run. To reflect the catching-up trends and driving factors of different economies in different sub-fields dynamically and comprehensively, we further tested the convergence of different economies in different sub-fields with the help of the β-convergence model, which is used in economic growth convergence analysis.

Convergence research is an important means of studying the distribution of economic growth between economies [10,11], and the most commonly used model is β-convergence. The β-convergence coefficient refers to the speed at which some latecomer indicators catch up with those of first movers. The so-called β-convergence means that if the latecomer grows faster than the first mover, the latecomer will catch up, and there is β-convergence. This study applies β-convergence of economic growth to the distribution of technological growth. The regression equation of β-convergence is:

$$\frac{1}{T} \log \frac{y_{i,\,t}}{y_{i,\,t-T}} = \alpha + \beta \log y_{i,\,t-T} + \mu_i$$

Furthermore, the regression equation of β-conditional convergence is:

$$\frac{1}{T} \log \frac{y_{i,\,t}}{y_{i,\,t-T}} = \alpha + \beta \log y_{i,\,t-T} + \beta_i X_i + \mu_i$$

where $i$ is economy; $t$ is time; $T$ is the length of the observation time; $y_{i,t}$ and $y_{i,\,t-T}$ represent the number of patents at the end and beginning of the observation period, respectively; and β is the convergence rate. $X_i$ represents the control variables.

Based on previous research, we take corruption (measured by control of corruption, legal environment (measured by rule of law), investment climate (measured by portfolio investment), trade (measured by merchandise trade of GDP), FDI (measured by foreign direct investment, net inflows), education (measured by mean years of schooling), financial development (measured by domestic credit to private sector of GDP), and absorptive capacity (measured by researchers in R&D per billion people) as control variables [61–67]. All the data were obtained from the World Bank (WDI and WGI) and UNESCO. Owing to the availability of data, the period was 1996–2014.

Table 3 shows the empirical results of β-convergence. It can be seen that eight economies have technological convergence in SOC, COM, DTV, SOL, MOB, and ROB. Asian economies and the United States are also experiencing technological convergence in SOC, COM, DTV, SOL, MOB, and ROB. However, there is no technological convergence between European economies and the US. Thus, it can be seen that Asian economies, as latecomers, tend to converge with the United States in the growth of technological capability through rapid technological development. Tables 4–6 show the regression results of conditional convergence with the control variables. Table 4 shows the convergence results for all economies, Table 5 shows the convergence results for the US and European economies, and Table 6 shows the convergence results for the US and Asian economies. Thus, we can conclude that conditional convergence exists among economies in SOL. Education level is an important driver of convergence between European economies and the United States, whereas the legal environment is an important factor in the convergence between Asian economies and the United States.

**Table 3.** Regression results based on β-convergence model.

| Economies | | SOC | COM | NET | ESW | DTV | SOL | RFID | MOB | ROB |
|---|---|---|---|---|---|---|---|---|---|---|
| All | β | −0.009 ** | −0.024 *** | 0.016 *** | −0.000 | −0.007 * | −0.021 ** | −0.002 | −0.021 *** | −0.009 * |
| | | (−2.28) | (−4.09) | (2.77) | (−0.00) | (−1.89) | (−2.29) | (−0.44) | (−3.79) | (−1.73) |
| | Constant | 0.209 *** | 0.271 *** | 0.008 | 0.085 *** | 0.141 *** | 0.344 *** | 0.133 *** | 0.215 *** | 0.098 *** |
| | | (9.15) | (6.76) | (0.26) | (4.06) | (5.77) | (7.17) | (4.26) | (6.26) | (4.09) |
| | Observations | 126 | 126 | 126 | 126 | 126 | 126 | 126 | 126 | 126 |
| | R-squared | 0.040 | 0.119 | 0.058 | 0.000 | 0.028 | 0.040 | 0.002 | 0.104 | 0.024 |
| US, European economies | β | 0.004 | 0.008 | 0.038 *** | 0.006 | 0.009 *** | −0.012 | 0.007 | 0.009 * | 0.013 * |
| | | (0.87) | (1.63) | (5.28) | (1.02) | (2.72) | (−1.13) | (1.00) | (1.91) | (1.70) |
| | Constant | 0.113 *** | 0.006 | −0.166 *** | 0.051 * | 0.006 | 0.269 *** | 0.032 | −0.008 | −0.017 |
| | | (4.53) | (0.17) | (−4.11) | (1.80) | (0.31) | (4.76) | (0.73) | (−0.27) | (−0.50) |
| | Observations | 72 | 72 | 72 | 72 | 72 | 72 | 72 | 72 | 72 |
| | R-squared | 0.011 | 0.036 | 0.285 | 0.015 | 0.096 | 0.018 | 0.014 | 0.050 | 0.040 |
| US, Asian economies | β | −0.017 *** | −0.031 *** | 0.005 | −0.003 | −0.016 *** | −0.024 ** | −0.008 | −0.028 *** | −0.017 *** |
| | | (−3.83) | (−4.56) | (0.92) | (−0.49) | (−3.82) | (−2.22) | (−1.51) | (−4.29) | (−3.16) |
| | Constant | 0.291 *** | 0.371 *** | 0.126 *** | 0.106 *** | 0.240 *** | 0.393 *** | 0.205 *** | 0.293 *** | 0.155 *** |
| | | (9.71) | (7.63) | (3.88) | (3.87) | (7.83) | (6.25) | (5.95) | (6.91) | (5.80) |
| | Observations | 72 | 72 | 72 | 72 | 72 | 72 | 72 | 72 | 72 |
| | R-squared | 0.174 | 0.229 | 0.012 | 0.003 | 0.172 | 0.066 | 0.031 | 0.208 | 0.125 |

Note: *** $p < 0.01$, ** $p < 0.05$, * $p < 0.1$.

**Table 4.** Regression results based on β-conditional convergence model among all economies.

| | SOC | COM | NET | ESW | DTV | SOL | RFID | MOB | ROB |
|---|---|---|---|---|---|---|---|---|---|
| β | 0.013 * | 0.016 | 0.029 *** | 0.018 | 0.031 *** | −0.083 *** | 0.029 *** | 0.002 | 0.007 |
| | (1.73) | (1.62) | (2.93) | (1.52) | (4.33) | (−4.61) | (2.80) | (0.13) | (0.60) |
| Corruption | −0.090 ** | −0.113 *** | −0.112 *** | −0.107 ** | 0.045 | −0.554 *** | −0.109 *** | 0.012 | −0.071 * |
| | (−2.50) | (−2.83) | (−2.64) | (−2.55) | (1.49) | (−7.80) | (−2.64) | (0.24) | (−1.79) |
| Legal environment | −0.014 | −0.155 *** | −0.095 * | 0.035 | −0.088 ** | 0.074 | 0.035 | −0.203 *** | −0.087 |
| | (−0.34) | (−2.81) | (−1.75) | (0.71) | (−2.49) | (0.76) | (0.64) | (−3.28) | (−1.63) |
| Investment climate | 0.000 | −0.000 | −0.000 | 0.000 | 0.000 | −0.000 * | 0.000 | −0.000 | −0.000 |
| | (0.36) | (−1.06) | (−0.33) | (0.62) | (1.54) | (−1.77) | (1.39) | (−0.02) | (−0.43) |
| Trade | −0.001 | −0.000 | 0.002 ** | 0.000 | 0.002 *** | −0.007 *** | 0.003 *** | 0.001 | −0.000 |
| | (−1.15) | (−0.37) | (2.19) | (0.02) | (2.81) | (−4.00) | (2.99) | (0.58) | (−0.30) |
| FDI | 0.000 | −0.000 | 0.000 | −0.000 | −0.000 | 0.000 | 0.000 | 0.000 | 0.000 |
| | (0.36) | (−0.19) | (1.33) | (−0.61) | (−0.01) | (1.10) | (0.45) | (1.10) | (0.07) |
| Education | 0.026 ** | 0.061 *** | 0.040 *** | 0.042 *** | −0.009 | 0.229 *** | 0.004 | 0.036 ** | 0.051 *** |
| | (2.19) | (4.98) | (3.08) | (3.06) | (−0.93) | (10.19) | (0.28) | (2.13) | (3.40) |
| Financial development | −0.001 *** | −0.001 *** | 0.000 | −0.001 | 0.000 | −0.003 *** | 0.000 | −0.000 | −0.001 * |
| | (−2.99) | (−2.82) | (0.03) | (−1.51) | (1.31) | (−3.80) | (0.62) | (−0.69) | (−1.88) |
| Absorptive capacity | −0.000 *** | −0.000 *** | −0.000 | −0.000 *** | −0.000 *** | −0.000 | −0.000 * | −0.000 | −0.000 ** |
| | (−2.74) | (−2.72) | (−0.82) | (−4.40) | (−3.28) | (−1.44) | (−1.85) | (−0.67) | (−2.44) |
| Constant | 0.222 *** | −0.067 | −0.332 *** | −0.079 | 0.063 | 0.393 *** | −0.033 | −0.052 | −0.112 |
| | (3.67) | (−0.86) | (−4.19) | (−1.09) | (1.25) | (6.25) | (−0.42) | (−0.58) | (−1.45) |
| Observations | 126 | 126 | 126 | 126 | 126 | 126 | 126 | 126 | 126 |
| R-squared | 0.520 | 0.588 | 0.574 | 0.347 | 0.566 | 0.577 | 0.451 | 0.409 | 0.350 |

Note: *** $p < 0.01$, ** $p < 0.05$, * $p < 0.1$.

**Table 5.** Regression results based on β-conditional convergence model among European economies and the US.

| | SOC | COM | NET | ESW | DTV | SOL | RFID | MOB | ROB |
|---|---|---|---|---|---|---|---|---|---|
| β | 0.017 | −0.029 *** | 0.009 | −0.013 | 0.017 * | −0.106 *** | 0.042 * | 0.009 | 0.001 |
| | (1.58) | (−2.92) | (0.44) | (−1.03) | (1.91) | (−6.20) | (1.79) | (0.72) | (0.05) |
| Corruption | 0.077 | 0.013 | 0.091 | −0.022 | 0.034 | −0.267 *** | 0.202 * | 0.099 * | 0.024 |
| | (1.21) | (0.37) | (0.93) | (−0.39) | (0.91) | (−2.73) | (2.00) | (1.68) | (0.25) |
| Legal environment | −0.185 | −0.136 | −0.167 | 0.192 * | 0.117 * | 0.165 | −0.060 | −0.199 * | −0.238 |
| | (−1.56) | (−1.54) | (−0.99) | (1.80) | (1.72) | (0.91) | (−0.34) | (−1.97) | (−1.48) |
| Investment climate | 0.000 | 0.000 | −0.000 | −0.000 | 0.000 | −0.000 | 0.000 | 0.000 | 0.000 |
| | (1.39) | (0.40) | (−0.02) | (−0.09) | (0.59) | (−1.41) | (1.42) | (0.85) | (0.41) |
| Trade | 0.001 | −0.001 | −0.000 | −0.004 *** | −0.001 | −0.014 *** | 0.005 ** | 0.002 | 0.001 |
| | (0.76) | (−0.92) | (−0.19) | (−2.67) | (−1.16) | (−5.29) | (2.14) | (1.36) | (0.55) |
| FDI | 0.000 | 0.000 | 0.000 | 0.000 | −0.000 | 0.000 | 0.000 | 0.000 | −0.000 |
| | (0.56) | (0.91) | (1.44) | (0.21) | (−0.09) | (1.06) | (1.33) | (1.22) | (−0.36) |
| Education | −0.002 | 0.047 *** | 0.019 | 0.045 *** | 0.001 | 0.204 *** | −0.055 ** | 0.014 | 0.037 |
| | (−0.08) | (5.92) | (0.76) | (2.82) | (0.09) | (8.43) | (−2.09) | (0.80) | (1.21) |
| Financial development | 0.000 | 0.001 | 0.000 | −0.001 | −0.001 | −0.003 *** | 0.001 | 0.000 | 0.001 |
| | (0.39) | (1.36) | (0.09) | (−1.65) | (−1.44) | (−2.72) | (1.51) | (0.72) | (0.69) |
| Absorptive capacity | −0.000 *** | −0.000 * | 0.000 *** | −0.000 *** | −0.000 * | −0.000 | −0.000 | −0.000 | −0.000 ** |
| | (−2.66) | (−1.71) | (2.86) | (−4.32) | (−1.72) | (−1.27) | (−0.78) | (−0.68) | (−2.37) |
| Constant | 0.357 *** | −0.079 | −0.502 *** | −0.109 | −0.090 | −0.583 *** | −0.019 | −0.127 | 0.042 |
| | (3.06) | (−0.95) | (−2.96) | (−1.00) | (−1.26) | (−3.18) | (−0.10) | (−1.21) | (0.26) |
| Observations | 72 | 72 | 72 | 72 | 72 | 72 | 72 | 72 | 72 |
| R-squared | 0.293 | 0.569 | 0.496 | 0.609 | 0.417 | 0.768 | 0.140 | 0.371 | 0.263 |

Note: *** $p < 0.01$, ** $p < 0.05$, * $p < 0.1$.

**Table 6.** Regression results based on β-conditional convergence model among Asian economies and the US.

| | SOC | COM | NET | ESW | DTV | SOL | RFID | MOB | ROB |
|---|---|---|---|---|---|---|---|---|---|
| β | 0.026 * | 0.037 * | 0.058 *** | 0.074 *** | 0.038 *** | −0.061 ** | 0.047 *** | 0.029 | 0.049 *** |
| | (1.82) | (1.91) | (5.42) | (3.31) | (2.90) | (−2.17) | (3.52) | (1.13) | (2.70) |
| Corruption | −0.065 | −0.070 | −0.131 ** | −0.108 * | −0.013 | −0.375 *** | −0.194 *** | −0.014 | −0.076 |
| | (−1.21) | (−0.84) | (−2.55) | (−1.78) | (−0.23) | (−3.14) | (−3.54) | (−0.14) | (−1.28) |
| Legal environment | 0.014 | 0.050 | −0.083 | 0.067 | −0.069 | 0.348 * | −0.022 | −0.051 | 0.049 |
| | (0.17) | (0.39) | (−1.07) | (0.75) | (−0.82) | (1.93) | (−0.25) | (−0.33) | (0.55) |
| Investment climate | 0.000 | −0.000 | 0.000 | 0.000 | 0.000 | −0.000 | 0.000 | −0.000 | −0.000 |
| | (0.57) | (−0.32) | (0.37) | (0.56) | (0.61) | (−0.02) | (0.44) | (−0.09) | (−0.47) |
| Trade | −0.001 | −0.002 | 0.002 ** | −0.000 | 0.001 | −0.002 | 0.002 * | −0.001 | −0.001 |
| | (−1.31) | (−0.95) | (2.18) | (−0.03) | (0.46) | (−1.01) | (1.67) | (−0.47) | (−0.39) |
| FDI | 0.000 | 0.000 | −0.000 * | −0.000 | 0.000 | 0.001 | −0.000 | 0.000 | 0.000 |
| | (0.50) | (0.42) | (−1.68) | (−0.96) | (0.39) | (1.51) | (−1.62) | (0.59) | (0.42) |
| Education | −0.002 | −0.044 | 0.050 * | −0.018 | −0.016 | 0.097 | 0.041 | −0.042 | −0.029 |
| | (−0.05) | (−0.86) | (1.71) | (−0.50) | (−0.50) | (1.49) | (1.26) | (−0.67) | (−0.88) |
| Financial development | −0.002 *** | −0.003 *** | 0.000 | −0.002 *** | −0.000 | −0.005 *** | 0.000 | −0.002 | −0.002 *** |
| | (−3.67) | (−3.44) | (0.31) | (−3.15) | (−0.25) | (−3.72) | (0.41) | (−1.39) | (−3.74) |
| Absorptive capacity | −0.000 | 0.000 | −0.000 *** | −0.000 | −0.000 | −0.000 | −0.000 *** | 0.000 | 0.000 |
| | (−1.00) | (0.37) | (−3.34) | (−1.21) | (−0.79) | (−0.18) | (−2.68) | (0.63) | (0.23) |
| Constant | 0.504 ** | 0.848 ** | −0.394 * | 0.397 | 0.176 | 0.275 | −0.285 | 0.604 | 0.531 ** |
| | (2.04) | (2.27) | (−1.75) | (1.47) | (0.73) | (0.53) | (−1.11) | (1.32) | (2.13) |
| Observations | 72 | 72 | 72 | 72 | 72 | 72 | 72 | 72 | 72 |
| R-squared | 0.630 | 0.593 | 0.700 | 0.528 | 0.534 | 0.700 | 0.674 | 0.383 | 0.533 |

Note: *** $p < 0.01$, ** $p < 0.05$, * $p < 0.1$.

## 6. Conclusions, Policy Implications, and Limitations

Based on TLC theory and logistic modeling, this study analyzed the catching-up characteristics of latecomer economies (with the US as the first mover), including take-off time, growth time, growth rate, and ceiling values. The results are summarized as follows.

First, along the take-off timeline, European economies and the US developed first, followed by Japan, Korea, and Taiwan, with China (Mainland) coming later. European economies have non-synchronous ICT development strategies, whereas Asian economies have synchronous ones. In terms of growth rate, Asian economies have a higher growth rate than European economies, showing a flying geese pattern.

Second, China (Mainland) has an increased growth rate from the first to the second process, and Korea and Taiwan have achieved an increase in several sub-fields in the indigenous process. Korea and Taiwan have prioritized strategies when selecting indigenous innovation sub-fields in the second process. In contrast, European economies are standing still from the first to the second process, and a few sub-fields are catching up in the second process.

Third, the technological growth rates of Asian economies are higher than those of the United States and European economies, and there is an obvious convergence with the United States. There is also an obvious catching up between Asian economies and the US in many sub-fields. In the field of SOL, European economies, Asian economies, and the United States have conditional convergence, among which education level is an important factor in the convergence between European economies and the United States, while laws and regulations are important factors in the convergence between Asian economies and the United States.

These findings have several policy implications for latecomers. Asian economies are catching up with Western economies. However, they should pay more attention to sustainability as they have a shorter growth time than Western economies. European economies did not fall behind and could still catch up. Like Korea and Taiwan, they may benefit from paying attention to advantageous sub-fields in the indigenous innovation process.

Some of the limitations inspire further research on this topic. First, this study did not point out the advantageous sub-fields to develop first in European economies in the second process. Further studies could aim to detect the comparative advantage index of each sub-field for European economies and choose the advantageous technology sub-fields as a priority development. Second, this study focused only on the catching-up characteristics in ICT. However, the extent to which these findings are valid for other fields has not yet been discussed. ICT is an enabling technology that can promote the development of other areas within their development trajectories. Thus, the findings might translate to a later take-off time for technology fields highly dependent on or related to ICT, as more time is required to apply and assimilate ICT in their areas. Other fields unrelated to, or enabled by, ICT may present completely different patterns. According to the resource-based view, if an economy focuses on specific fields, it may have fewer resources to develop other fields. Further research should examine catching-up characteristics and timing in other sub-fields related and unrelated to ICT.

**Author Contributions:** Conceptualization, N.Z.; methodology, C.S.; software, C.S. and M.X.; validation, X.W. and J.D.; data curation, C.S. and J.D.; writing—original draft preparation, N.Z.; writing—review and editing, N.Z., C.S. and M.X.; visualization, N.Z.; supervision, N.Z. All authors have read and agreed to the published version of the manuscript.

**Funding:** This research was funded by a program of the National Natural Science Foundation of China, grant numbers 72002021 and 42030409, and the China Postdoctoral Science Foundation, grant number 2021M690499.

**Institutional Review Board Statement:** Not applicable.

**Informed Consent Statement:** Not applicable.

**Data Availability Statement:** Not applicable.

**Conflicts of Interest:** The authors declare no conflict of interest.

## Appendix A

**Table A1.** The take-Off time for each economy in nine sub-fields (simple logistic model).

| Take-Off Time | | SOC | COM | NET | ESW | DTV | SOL | RFID | MOB | ROB | Standard Deviation |
|---|---|---|---|---|---|---|---|---|---|---|---|
| United States | $T_{0.1}$ rank | 1997 3 | 1995 5 | 2007 6 | 1987 5 | 1992 4 | 2008 5 | 2000 4 | 1983 3 | 1968 2 | 11.79 |
| Great Britain | $T_{0.1}$ rank | 1986 1 | 1971 1 | 2001 3 | 1975 2 | 1978 1 | 1998 1 | 1984 2 | 1972 1 | 1958 1 | 12.80 |
| France | $T_{0.1}$ rank | 2001 5 | 1979 2 | 1989 1 | 1971 1 | 1978 1 | 2004 4 | 1980 1 | 1977 2 | 1971 3 | 11.42 |
| Germany | $T_{0.1}$ rank | 2001 5 | 1984 3 | 1999 2 | 1979 3 | 1980 3 | 2001 3 | 1991 3 | 1990 5 | 1974 4 | 9.57 |
| Japan | $T_{0.1}$ rank | 1996 2 | 1991 4 | 2001 3 | 1986 4 | 1997 5 | 2000 2 | 2001 5 | 1988 4 | 1982 5 | 6.65 |
| Korea | $T_{0.1}$ rank | 2002 7 | 2004 7 | 2007 6 | 1999 6 | 2008 6 | 2010 6 | 2007 7 | 2006 6 | 1996 6 | 4.29 |
| Taiwan | $T_{0.1}$ rank | 1999 4 | 2003 6 | 2006 5 | 2002 7 | 2008 6 | 2010 6 | 2006 6 | 2007 7 | 2005 7 | 3.14 |
| China (Mainland) | $T_{0.1}$ rank | 2012 8 | 2009 8 | 2010 8 | 2011 8 | 2010 8 | 2011 8 | 2010 8 | 2010 8 | 2008 8 | 1.10 |

Note: Ranks are based on take-off time of eight economies in each sub-field.

**Table A2.** The growth rate and time for each economy in nine sub-fields (simple logistic model).

| Growth Rate (Growth Time) | | SOC | COM | NET | ESW | DTV | SOL | RFID | MOB | ROB | Avg. |
|---|---|---|---|---|---|---|---|---|---|---|---|
| United States | $b$ | 0.13 (34) | 0.08 (54) | 0.15 (30) | 0.08 (54) | 0.08 (54) | 0.18 (25) | 0.08 (54) | 0.08 (54) | 0.08 (53) | 0.11 (46) |
| Great Britain | $b$ | 0.08 (54) | 0.08 (54) | 0.09 (47) | 0.08 (53) | 0.08 (53) | 0.18 (25) | 0.08 (53) | 0.08 (53) | 0.08 (54) | 0.09 (50) |
| France | $b$ | 0.08 (52) | 0.08 (53) | 0.08 (53) | 0.08 (53) | 0.10 (46) | 0.12 (38) | 0.08 (53) | 0.08 (54) | 0.08 (54) | 0.09 (51) |
| Germany | $b$ | 0.09 (51) | 0.08 (53) | 0.09 (52) | 0.08 (54) | 0.08 (53) | 0.16 (27) | 0.08 (53) | 0.08 (53) | 0.08 (54) | 0.09 (50) |
| Japan | $b$ | 0.16 (28) | 0.13 (35) | 0.13 (34) | 0.14 (32) | 0.12 (38) | 0.16 (28) | 0.11 (41) | 0.12 (36) | 0.11 (39) | 0.13 (35) |
| Korea | $b$ | 0.23 (20) | 0.19 (23) | 0.27 (16) | 0.20 (22) | 0.18 (24) | 0.29 (15) | 0.26 (17) | 0.16 (28) | 0.17 (26) | 0.22 (21) |
| Taiwan | $b$ | 0.25 (18) | 0.29 (15) | 0.41 (11) | 0.33 (13) | 0.32 (14) | 0.29 (15) | 0.37 (12) | 0.34 (13) | 0.29 (15) | 0.32 (14) |
| China (Mainland) | $b$ | 0.55 (8) | 0.59 (8) | 0.66 (7) | 0.40 (11) | 0.65 (7) | 0.70 (6) | 0.75 (6) | 0.47 (9) | 0.54 (8) | 0.59 (8) |

Note: Growth times are in parentheses. $b$ is the growth rate.

**Table A3.** The ceiling values for each economy in nine sub-fields (simple logistic model).

| Ceiling Value | | SOC | COM | NET | ESW | DTV | SOL | RFID | MOB | ROB |
|---|---|---|---|---|---|---|---|---|---|---|
| United States | $K$ | 251,748 | 703,141 | 331,802 | 66,859 | 385,901 | 527,474 | 372,802 | 281,311 | 31,309 |
| | rank | 1 | 1 | 1 | 1 | 1 | 1 | 1 | 1 | 1 |
| Great Britain | $K$ | 3248 | 7011 | 9577 | 1631 | 10,034 | 2854 | 7147 | 7889 | 1268 |
| | rank | 8 | 7 | 4 | 4 | 6 | 5 | 5 | 6 | 4 |
| France | $K$ | 8898 | 8671 | 3894 | 971 | 5469 | 3064 | 4988 | 8274 | 1226 |
| | rank | 5 | 5 | 7 | 6 | 7 | 4 | 6 | 5 | 5 |
| Germany | $K$ | 18,540 | 21,302 | 9145 | 1842 | 10,852 | 2378 | 10,521 | 27,383 | 4124 |
| | rank | 4 | 4 | 5 | 3 | 5 | 6 | 4 | 3 | 3 |
| Japan | $K$ | 69,438 | 105,154 | 41,399 | 9300 | 223,354 | 32,551 | 72,285 | 70,100 | 13,327 |
| | rank | 2 | 2 | 2 | 2 | 2 | 2 | 2 | 2 | 2 |
| Korea | $K$ | 23,771 | 33,806 | 14,327 | 1621 | 84,841 | 8067 | 18,938 | 25,475 | 1193 |
| | rank | 3 | 3 | 3 | 5 | 3 | 3 | 3 | 4 | 6 |
| Taiwan | $K$ | 5750 | 8395 | 933 | 617 | 16,104 | 1839 | 2817 | 3971 | 469 |
| | rank | 6 | 6 | 8 | 8 | 4 | 7 | 8 | 7 | 7 |
| China (Mainland) | $K$ | 5013 | 6207 | 4758 | 713 | 4916 | 1451 | 3126 | 2701 | 410 |
| | rank | 7 | 8 | 6 | 7 | 8 | 8 | 7 | 8 | 8 |

## Appendix B

**Table A4.** The take-off time in the first and second processes for each economy in nine sub-fields.

| Ceiling Value | | SOC | COM | NET | ESW | DTV | SOL | RFID | MOB | ROB |
|---|---|---|---|---|---|---|---|---|---|---|
| United States | FC | 1965 | 1962 | 1974 | 1961 | 1967 | 1977 | 1964 | 1966 | 1962 |
| | SC | 1997 | 1993 | 2002 | 1995 | 1993 | 2005 | 1997 | 1993 | 1993 |
| Great Britain | FC | 1975 | 1968 | 1975 | 1965 | 1974 | 1978 | 1965 | 1971 | 1965 |
| | SC | 2001 | 1993 | 2003 | 1989 | 1997 | 2004 | 1993 | 2000 | 1990 |
| France | FC | 1970 | 1967 | 1974 | 1966 | 1970 | 1976 | 1969 | 1969 | 1966 |
| | SC | 2002 | 1996 | 2000 | 1996 | 1999 | 2006 | 1998 | 1997 | 1992 |
| Germany | FC | 1975 | 1978 | 1976 | 1977 | 1975 | 1979 | 1977 | 1978 | 1978 |
| | SC | 2003 | 1999 | 2005 | 1995 | 2003 | 2005 | 2003 | 1997 | 1994 |
| Japan | FC | 1997 | 1994 | 1996 | 1996 | 1994 | 1998 | 1983 | 1994 | 1991 |
| | SC | 2007 | 2007 | 2010 | 2006 | 2008 | 2011 | 2010 | 2008 | 1990 |
| Korea | FC | 1999 | 1997 | 2004 | 2002 | 1992 | 1997 | 2004 | 1993 | 2002 |
| | SC | 2008 | 2008 | 2009 | 2009 | 2010 | 2011 | 2010 | 2010 | 2009 |
| Taiwan | FC | 1996 | 1995 | 2006 | 1997 | 1996 | 1998 | 1996 | 1996 | 1997 |
| | SC | 2013 | 2012 | 2013 | 2013 | 2012 | 2013 | 2012 | 2012 | 2011 |
| China (Mainland) | FC | 1965 | 1962 | 1974 | 1961 | 1967 | 1977 | 1964 | 1966 | 1962 |
| | SC | 1997 | 1993 | 2002 | 1995 | 1993 | 2005 | 1997 | 1993 | 1993 |

Note: FC and SC represent the first and second curves, respectively.

**Table A5.** The growth rate and growth time in two processes for each economy in nine sub-fields.

| Growth Rate (Growth Time) | | SOC | COM | NET | ESW | DTV | SOL | RFID | MOB | ROB |
|---|---|---|---|---|---|---|---|---|---|---|
| Great Britain | $d$ | 0.18 (24) | 0.17 (26) | 0.17 (26) | 0.18 (25) | 0.17 (26) | 0.19 (23) | 0.17 (26) | 0.17 (26) | 0.18 (24) |
| | $f$ | 0.21 (21) | 0.17 (26) | 0.25 (18) | 0.19 (23) | 0.18 (25) | 0.45 (10) | 0.18 (24) | 0.18 (25) | 0.17 (25) |
| France | $d$ | 0.18 (25) | 0.17 (27) | 0.18 (25) | 0.21 (21) | 0.17 (26) | 0.2 (22) | 0.17 (26) | 0.17 (26) | 0.2 (23) |
| | $f$ | 0.18 (24) | 0.18 (25) | 0.18 (25) | 0.17 (25) | 0.18 (24) | 0.35 (13) | 0.17 (26) | 0.20 (22) | 0.18 (25) |
| Germany | $d$ | 0.17 (26) | 0.17 (26) | 0.18 (25) | 0.17 (25) | 0.17 (26) | 0.18 (24) | 0.18 (25) | 0.17 (26) | 0.19 (23) |
| | $f$ | 0.28 (16) | 0.18 (24) | 0.21 (21) | 0.24 (19) | 0.18 (25) | 0.44 (10) | 0.24 (19) | 0.19 (23) | 0.17 (26) |
| Japan | $d$ | 0.18 (25) | 0.23 (19) | 0.20 (22) | 0.21 (21) | 0.18 (25) | 0.19 (23) | 0.21 (21) | 0.24 (18) | 0.23 (19) |
| | $f$ | 0.32 (14) | 0.25 (18) | 0.32 (14) | 0.22 (20) | 0.27 (16) | 0.41 (11) | 0.30 (15) | 0.20 (22) | 0.18 (24) |
| Korea | $d$ | 0.51 (9) | 0.41 (11) | 0.39 (11) | 0.44 (10) | 0.35 (13) | 0.45 (10) | 0.21 (21) | 0.45 (10) | 0.33 (13) |
| | $f$ | 0.48 (9) | 0.45 (10) | 0.74 (6) | 0.36 (12) | 0.51 (9) | 0.75 (6) | 0.68 (6) | 0.40 (11) | 0.34 (13) |
| Taiwan | $d$ | 0.58 (8) | 0.51 (9) | 0.72 (6) | 0.58 (8) | 0.31 (14) | 0.42 (10) | 0.78 (6) | 0.32 (14) | 0.59 (7) |
| | $f$ | 0.51 (9) | 0.55 (8) | 0.66 (7) | 0.67 (7) | 0.78 (6) | 0.79 (6) | 0.61 (7) | 0.75 (6) | 0.64 (7) |
| China (Mainland) | $d$ | 0.35 (12) | 0.34 (13) | 0.81 (5) | 0.41 (11) | 0.36 (12) | 0.43 (10) | 0.38 (12) | 0.35 (13) | 0.51 (9) |
| | $f$ | 1.46 (3) | 1.35 (3) | 1.53 (3) | 0.95 (5) | 1.55 (3) | 1.19 (4) | 1.49 (3) | 1.33 (3) | 1.17 (4) |

Note: $d$ and $f$ are the growth rates of the first and second curves, respectively. Growth times are in parentheses. $d$ is growth rate for the first process; $f$ is growth rate for the second process.

**Table A6.** The ceiling values in the first and second processes for each economy in nine sub-fields.

| Ceiling Value | | SOC | COM | NET | ESW | DTV | SOL | RFID | MOB | ROB |
|---|---|---|---|---|---|---|---|---|---|---|
| Great Britain | $K_1$ | 437 | 2176 | 681 | 407 | 2271 | 471 | 1119 | 2473 | 744 |
| | $K_2$ | 1547 | 4393 | 2600 | 1015 | 5338 | 1426 | 4246 | 4319 | 580 |
| France | $K_1$ | 588 | 1963 | 716 | 252 | 2226 | 205 | 947 | 2827 | 376 |
| | $K_2$ | 2947 | 4388 | 2441 | 560 | 3006 | 668 | 2806 | 4831 | 659 |
| Germany | $K_1$ | 1248 | 3571 | 673 | 428 | 2864 | 303 | 1193 | 3443 | 1090 |
| | $K_2$ | 4067 | 11,297 | 2577 | 871 | 7116 | 1032 | 3733 | 11,656 | 2366 |
| Japan | $K_1$ | 11,672 | 18,791 | 4082 | 2052 | 26,808 | 4277 | 5863 | 12,507 | 3931 |
| | $K_2$ | 37,755 | 59,095 | 15,613 | 6255 | 94,832 | 13,415 | 19,526 | 43,378 | 8852 |
| Korea | $K_1$ | 4022 | 3899 | 1665 | 345 | 5526 | 574 | 1949 | 2006 | 366 |
| | $K_2$ | 12,641 | 14,354 | 5301 | 982 | 20,954 | 2337 | 7363 | 8454 | 801 |
| Taiwan | $K_1$ | 2191 | 1571 | 187 | 280 | 1894 | 129 | 587 | 445 | 100 |
| | $K_2$ | 3297 | 5295 | 589 | 309 | 7072 | 606 | 1817 | 1628 | 262 |
| China (Mainland) | $K_1$ | 325 | 1085 | 635 | 51.2 | 737 | 209 | 572 | 360 | 63.9 |
| | $K_2$ | 1592 | 3376 | 2326 | 305 | 2438 | 1049 | 1928 | 1172 | 235 |

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
