# Peer review of "Catching Up of Latecomer Economies in ICT for Sustainable Development: An Analysis Based on Technology Life Cycle Using Patent Data"

_sustainability, doi:10.3390/su15119038_

Round 1

Reviewer 1 Report

General Comments:

This paper discusses an important and relevant question regarding catching-up of one group of countries in Asia with respect to technologies of the other three selected groups of more developing countries from different geographical territories (viz., USA, Western Europe and Japan), chosen on the basis of their performances during last several years.  It is done for the period spanning 1960—2014, using patent data from USPTO.

The authors use Technology Life Cycle (TLC) and patent data analysis for this purpose.

Results of the study are: technical efficiency enhancement can close the technology gap between highly performing groups vis-à-vis the low performing nations with synchronous and non-synchronous way. Asian economies typically showed flying-geese pattern.

Degrees or extend differ across the sample.

The results are not novel or counter-intuitive, and it follows from the past literatures in other contexts. Thus, I can’t say that it’s novel in value-addition to research. Sub-sectors are new things to add.

However, it’s good in terms of creating new evidences as heaps of applied empirical works are doing these days.

Organization of documents needs lots of change in terms of presentation. Conclusion is presented in 5 and 6 sections twice. WHY?

What are the lists of variables and discussion of determinants for sub-field convergence is missing. Enlist all the variables in an equation or tabular format with intuitive explanation as to why they affect TLC and convergence?

Followings are important to be considered:-

(i) institutions (Corruption, Rule of Law, Investment Climate) are inadequate and of bad quality, (ii) insufficient amount of trade and FDI obstruct technology flows, and also (iii) lack of education, financial development, and inability to absorb frontier technology widens the gap between actual (effective) ad potential.

As stochastic frontier for regional group is used as necessary factors for technology catch up, which is defined in terms of metatechnology considered at the global level, meta frontier analysis is essential for tracing catch up and closing the gap. The author should emphasize this aspect and the necessity of microeconomic and macroeconomic management for an effective utilization of technological advancement.

Thus, the author needs to address several points before the paper is accepted for final publication. Below I present several specific comments. Thus, it is subject to some revision.

Specific Comments:

1) Needs discussion on some aspects of Barro-Convergence at sectoral level, Solow and TFP. If the author wants to make a case for distinguishing the current work with that of Solow, it needs to be bit explicit, at least, adding a footnote.

2) Section 1 mentions some of the literatures and the author also mentions some of these points while discussing the results in sections 4/5. However, the discussion in this section on existing studies and gap is loose and hence, it needs some elucidation. Also, some of the important literature is missing; for example, at least few like Aghion and Howitt, Chad Jones, R. E. Lucas etc.

In the same vein, it is helpful for the reader if the author spells out very briefly about new endogenous growth theory to explain productivity terms and technological progress. In fact, roles of both might be important and it’s a matter of degree.

3) It is better to be more explicit in rationalizing why estimation of economic/productive efficiency is better than just doing logistic calibration. As this paragraph claims the ‘main contribution’ of the paper, it is worthwhile to highlight clearly the distinction from the existing literature studying the same.

4) In 3rd section, the author needs to clear the concepts of and differences between terminologies such as: ‘Technological capability’, ‘Technical Efficiency (TE)’, Allocative efficiency, and Productive efficiency as these terms are used often in the text.

5) Author/s need to explain more on the theoretical background and explain equations in more detail with intuition to relate to the results that follow subsequently.

6) It is helpful to discuss distinction between TE relative to meta function and stochastic frontier while explaining the core equations.

7) It is better to give more rationale behind selection of countries in each group across geographies. Stylized facts are important as background motivation. Literature reference is inadequate and needs to be more particular for discussing different factors and their roles in explaining the technology gap ratio estimation.

Thus, the paper is interesting and covers an important area. It surely adds some value to the literature. Therefore, I recommend major revision along the lines before further resubmission for publication. 

see above.

Reviewer 2 Report

The study "Catching up of latecomer economies in ICT for sustainable development: An analysis based on technology life cycle analysis using patent data" was presented with excellent academic rigor and sound methodology was followed. The manuscript was easy to read and follow and could meaningfully contribute to scholarship. 

However, the authors must pay attention to the comments included in the reviewed and attached manuscript to improve their work. 

The layout of the manuscript could be improved by paying attention to section 5 and 6, Figure 2 and Figure 3. 

Pay attention to minor grammar mistakes like in line 52 - they must be changed to their. Revise the whole document and correct grammar accordingly.  

Reviewer 3 Report

1) Overall, there are a lot of grammatical mistakes that need to be taken care of. The authors should carefully go through the manuscript.

For example: In the Abstract Section, Using technology life cycle analysis and patenting data should be changed to "Using technology life cycle analysis and patent data"

The following sentence is also grammatically wrong: "Being at the leading edge 33 of ICT development and innovation has always been an objective for governments."

2) The title of this manuscript should be modified as follows: "Catching up of Latecomer Economies in ICT for Sustainable Development: An Analysis Based on Technology Life Cycle Using Patent Data"

3) In Section 3.1, please add a reference for the equation 5. 

4) In Section 3.2, if you choose only one database, you will naturally have a home-bias effect. But the authors described this in a opposite way. You rather want to use USPTO, as the most important innovations can be covered through this database. Please have a look at other publications and correct this point.

5) I don't fully understand what the authors want to express with the following sentence (please elaborate more on this): "even though the US has the largest number of patents in USPTO, 229 it also has many patents in other patent offices, that is, the US also has advantages in other 230 databases; in addition, the US is just a benchmark in this study" 

6) In Section 4.2, please explain more in detail how these two catching-up processes might differ. What was the focus of the first process ? Was there a change in context in the second catching-up process? How can you reasonably argue the difference between these two processes?

7) If you used Editage for English editing, they should re-do it for the revised manuscript.

8) Please add a reference (if there is any) to the explanation of the flying geese paradigm.

9) Please also discuss whether simply looking at the number of patents might actually reveal the national technologcial competitiveness.

The language (especially grammatical errors) needs to be improved.

Round 2

Reviewer 1 Report

Thank you for addressing most of the points and now the current draft reads much better although some more improvements can always be done. Buit now it is in good shape to be passed.

Thanks and best wishes